# Establishing coherent momentum-space electronic states in locally ordered materials

Samuel T. Ciocys[1,2,11], Quentin Marsal [3,4,11], Paul Corbae [2,5,11], Daniel Varjas [6,7,8,9], Ellis Kennedy [5,10], Mary Scott [5,10], Frances Hellman [1,2], Adolfo G. Grushin [3] & Alessandra Lanzara [1,2] ✉

Rich momentum-dependent electronic structure naturally arises in solids with long-range crystalline symmetry. Reliable and scalable quantum technologies rely on materials that are either not perfect crystals or non-crystalline, breaking translational symmetry. This poses the fundamental questions of whether coherent momentum-dependent electronic states can arise without long-range order, and how they can be characterized. Here we investigate $Bi_2Se_3$, which exists in crystalline, nanocrystalline, and amorphous forms, allowing direct comparisons between varying degrees of spatial ordering. Through angle-resolved photoemission spectroscopy, we show for the first time momentum-dependent band structure with Fermi surface repetitions in an amorphous solid. The experimental data is complemented by a model that accurately reproduces the vertical, dispersive features as well as the replication at higher momenta in the amorphous form. These results reveal that well-defined real-space length scales are sufficient to produce dispersive band structures, and that photoemission can expose the imprint of these length scales on the electronic structure.

Electronic coherence stands as a paramount consideration for harnessing and manipulating the quantum-mechanical properties of materials. The foundation of our understanding of solid-state physics utilized by modern technology is the realization that long-range order, in the form of discrete translational symmetry of the crystal lattice, is fundamental in establishing highly coherent electronic states in momentum-space. Translational symmetry, inherent in crystalline solids, leads to well-defined peaks in the lattice structure factor, as depicted in Fig. 1a top. These peaks dictate the locations of Bragg scattering of electrons, with scattering planes determined by the periodic atomic potential, outlining the edges of the Brillouin zone. Bloch's theorem, applied to crystalline solids, precisely characterizes energy- and momentum-dependent electronic states confined within periodic Brillouin zones in momentum space. These electronic states, delineated by crystalline momentum, ultimately define the ground state of materials and influence their transport, optical, magnetic, and topological properties[1].

In contrast, non-crystalline solids jeopardize the Brillouin zone description due to the absence of long-range order. For instance, the structure factor of random atomic positions is uniformly distributed, as illustrated in Fig. 1a middle. Lattice disorder is expected to localize electronic states, resulting in a structureless dispersion relation (i.e., a featureless momentum space)[2–4].

[1]Department of Physics, University of California, Berkeley, CA 94720, USA. [2]Materials Science Division, Lawrence Berkeley National Laboratory, Berkeley, CA 94720, USA. [3]Univ. Grenoble Alpes, CNRS, Grenoble INP, Institut Néel, 38000 Grenoble, France. [4]Department of Physics and Astronomy, Uppsala University, Box 516, 751 20 Uppsala, Sweden. [5]Department of Materials Science, University of California, Berkeley, CA 94720, USA. [6]Department of Physics, Stockholm University, AlbaNova University Center, 114 21 Stockholm, Sweden. [7]The Max Planck Institute for the Physics of Complex Systems, 01187 Dresden, Germany. [8]Department of Theoretical Physics, Institute of Physics, Budapest University of Technology and Economics, Műegyetem rkp. 3., H-1111, Budapest, Hungary. [9]IFW Dresden and Würzburg-Dresden Cluster of Excellence ct.qmat, Helmholtzstrasse 20, 01069 Dresden, Germany. [10]Molecular Foundry, Lawrence Berkeley National Laboratory, Berkeley, CA 94720, USA. [11]These authors contributed equally: Samuel T. Ciocys, Quentin Marsal, Paul Corbae. ✉e-mail: ale.lanzara@berkeley.edu

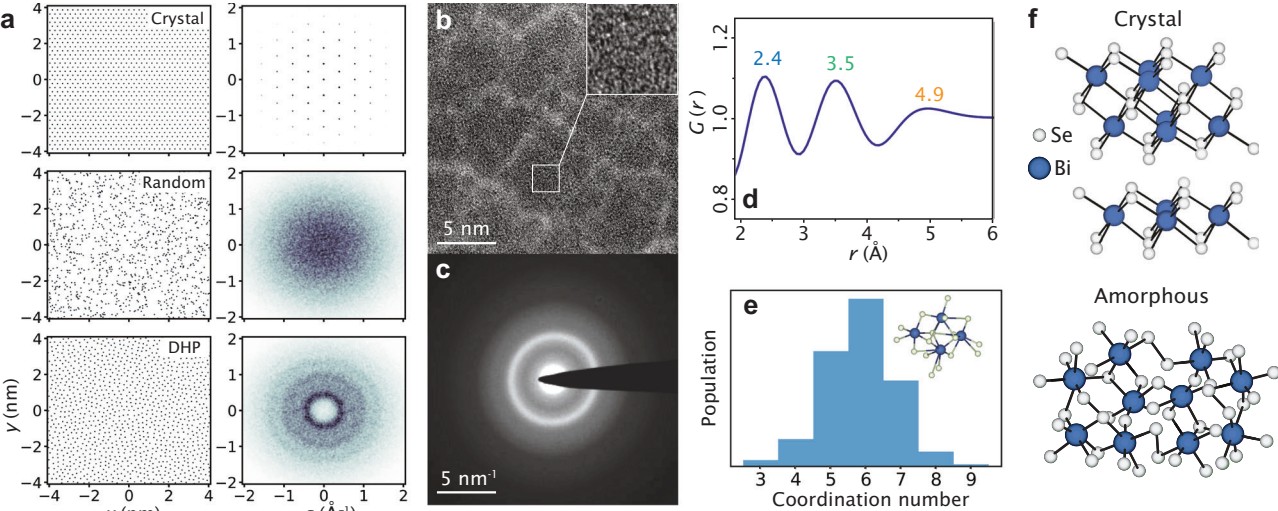

**Fig. 1 | Well-defined reciprocal length scale from real-space short-range order.** **a** Fourier transforms for three real-space point distributions (crystalline, normal random, and disordered hard pack) demonstrates that reciprocal-space structure persists in the presence of well defined nearest-neighbor distance. **b, c** Large scale HRTEM image shows no signs of crystalline order like precursor lattice fringes. The electron diffraction pattern shows broad diffuse rings corresponding to SRO and no high intensity spots from long range order. **d** The reduced radial distribution function, $G(r)$, has three peaks from a well defined nearest neighbor (2.4 Å), next nearest neighbor (3.5 Å), and third nearest neighbor (4.9 Å). **e** Coordination number for amorphous $Bi_2Se_3$ calculated using a 200 atom cell and ab-initio molecular dynamics, showing a peak at 6. Inset: an example coordination environment in amorphous $Bi_2Se_3$. **f** Ball-and-stick model of crystalline and amorphous $Bi_2Se_3$. For the amorphous structure, van der Waals separation is absent and majority sites are octahedral coordinated, implying an isotropic nearest neighbor distance.

Much of our world is composed of materials falling between these two extremes, including amorphous materials, high-entropy alloys, quasicrystals, and liquid metals[5–8]. Despite being aperiodic and typically lacking long-range order, these materials retain short-range ordering (SRO) with well-defined structural length scales, such as bond lengths and preferred local environments. Structurally, this situation results in an atomic arrangement that is locally similar to the crystalline case (bond lengths, angles, and coordination), but globally, the atomic sites demonstrate no periodic behavior[9–12]. In this case, the diffraction pattern is not uniformly distributed but presents a set of rings, as depicted in Fig. 1a bottom, each corresponding to characteristic real-space scales, such as a well-defined nearest neighbor distance.

Despite their central role in modern technologies, the extent to which atomic structure determines the electronic properties of amorphous materials remains an outstanding problem in condensed matter physics. Previous efforts to develop a theory for the electronic structure of liquid metals in the 1960s were primarily aimed at explaining X-ray diffraction results of the time, where dispersive electronic features were predicted to persist, contingent on hard-packing atomic arrangements[13–21]. Decades since, a direct band structure measurement of a liquid or an amorphous material, paired with a theoretical interpretation, is absent from the literature - a key criterion for well-understood material systems. While studies have utilized liquid metal theory to describe the interaction of disordered but spatially correlated dopants on a crystalline lattice[22] or accounted for the effect of disorder on the band structure of polycrystalline materials[23], they have not focused on purely amorphous atomic arrangements. The field has perhaps been hindered by the complexity of analyzing glassy configurations. Only in 2020 was the accurate determination of the microscopic atomic structure of an amorphous equivalent of graphene[24], demonstrating that the commonly assumed decades-old Zachariasen random-network picture cannot explain the distribution of atoms in amorphous graphene. However, grasping order in the vast majority of amorphous material is much harder, as they are typically three-dimensional or have multiple elements. This problem calls to develop further experimental tools and models to extract information on the order at the short and medium scale.

For instance, the recent experimental observation of strongly dispersive surface states in a purely amorphous $Bi_2Se_3$ (a-$Bi_2Se_3$)[25] challenges the necessity of a Bloch theorem foundation for coherent electronic structure in momentum space and demonstrates the need for predicting electronic behavior in materials lacking long-range order. The surface states have topological origin, but they are significantly more dispersive compared to those in the crystal, a counter-intuitive observation which remains unexplained.

In this study we present the first direct measurement of highly dispersive electronic states in an amorphous solid. These states form a distinct Fermi surface with band repetitions analogous to Brillouin zones and striking broad vertical features divergent from the crystalline case. Furthermore, we reveal a theoretical model that goes beyond liquid metal theory to describe previously unpredicted band renormalizations using amorphous Hamiltonians and scattering theory. Our theory explains the significant dispersion differences between crystalline and amorphous systems as rooted in scattering via well-defined length scales as seen in the amorphous structure factor. These results assert that, even in the absence of long-range order, a well-defined real-space length scale is sufficient to produce strongly dispersive band structures in amorphous matter. Revived attention should be placed on this broad class of materials for discovering and understanding novel momentum-dependent quantum phenomena, such as momentum pairing and spin-orbit coupling.

## Results

We begin by recalling that crystalline $Bi_2Se_3$ (c-$Bi_2Se_3$) features a quintuple layer structure of alternating selenium and bismuth planes with bismuth atoms octohedrally coordinated with six adjacent selenium atoms. The quintuple layers are bonded by van der Waals forces, the stacking of which defines the c-axis lattice constant (see top panel in Fig. 1f).

The accumulated knowledge on amorphous systems suggests that a realizable structure for a-$Bi_2Se_3$ can share certain traits with c-$Bi_2Se_3$. For example, in elemental amorphous materials, such as Si, Ge and monolayer carbon, or bi-elemental amorphous compounds such as $SiO_2$ and GaAs, the coordination of atoms and the nearest neighbor

distances remain peaked at the values of their crystalline counterparts. The structural disorder stems from small variations in bond angles and smaller variations in bond lengths, which are peaked at the crystalline values[12,24,26,27]. Following suit, a-$Bi_2Se_3$ is expected to also possess octohedrally coordinated bismuth atoms and similar local environment to c-$Bi_2Se_3$. The propensity for amorphous systems to retain the crystalline local order means that the amorphous system has a tendency to retain a well defined length-scale. A notable difference between the c-$Bi_2Se_3$ and a-$Bi_2Se_3$ however is that the van der Waals gap in c-$Bi_2Se_3$ is an inherently 2D structure with no obvious analog in the amorphous case. Indeed, in ref. 25 we have demonstrated through Raman spectroscopy that the van der Waals gap no longer exists in a-$Bi_2Se_3$. Using ab-initio molecular dynamics to generate realistic a-$Bi_2Se_3$ amorphous structures, we observe a peak in the coordination number at six (Fig. 1e), representing the existence of majority octahedral environments.

To elucidate the real-space structure, we grew a-$Bi_2Se_3$ using physical vapor deposition from two elemental effusion cells and characterized the structure using high-resolution transmission electron microscopy (HRTEM) in shown in Fig. 1b. The large scale HRTEM image indicates no regions exhibiting crystalline order or even nanocrystalline precursors (the contrast visible in the main image is associated with columnar microstructure that is common in thermally evaporated amorphous materials). The inset displays an expanded 2 nm x 2 nm field of view displaying a pattern due to phase contrast resulting from the lack of long-range periodicity, but has no sign of any nanocrystalline or even precursor nanocrystallites.

Panel (c) in Fig. 1 shows TEM diffraction from the same film exhibiting the characteristic diffuse rings of amorphous systems lacking long range order. The presence of rings is indicative of well-preserved real-space length-scales. Using parallel-beam diffraction, we compute the reduced radial distribution function, $G(r)$, shown in Fig. 1d. The a-$Bi_2Se_3$ film shows clear peaks at 2.4 Å, 3.5 Å, and 4.9 Å, indicating well-defined real and reciprocal length-scales in the system. Further data and detailed analysis regarding the structure of a-$Bi_2Se_3$, including X-ray diffraction and decapping, has been presented in reference[25].

Figure 2 summarizes the momentum-space structure from the electronic dispersion we obtain from ARPES on a-$Bi_2Se_3$. For comparison with the amorphous spectrum in panels (b)-(h), panel (a) displays ARPES spectra for c-$Bi_2Se_3$ and nanocrystalline $Bi_2Se_3$ at photon energies of 115 eV and 100 eV, respectively.

The crystalline sample exhibits a Dirac surface state and increased spectral intensity at the valence band near $E - E_F = -0.6$ eV. The nanocrystalline sample is momentum-independent, with the only energy-dependent feature above −1.0 eV being reduced spectral intensity at $E - E_F = -0.4$ eV.

Panel (b) displays the a-$Bi_2Se_3$ spectrum along a momentum-slice cutting through $(k_x, k_y) = (0, 0)$ at a photon energy of 120 eV, revealing remarkably dispersive band structure manifesting as vertical column-like features and an M-shaped valence band[25]. The near-vertical features are in stark contrast to the expectation that disorder and localization lead to a broadened, momentum-independent electronic dispersion[2–4], as seen in the nanocrystalline case. The clear difference between the amorphous and nanocrystalline spectra highlights how important the local environment is for electronic properties, with the possibility that grain boundaries are playing a substantial role in electronic decoherence.

Looking further at larger momenta, the band structure is replicated, resulting in copies of the electronic states at −1.75 and 1.75 Å⁻¹ with reduced intensity. The replicas occur at a characteristic momentum, $k^* = 2\pi/a^* \approx 1.75$ Å⁻¹ corresponding to $a^* \approx 3.6$ Å, which closely matches the second peak in the radial distribution function from Fig. 1d. The consequent Fermi surface can be seen in Fig. 2c, confirming a rotationally symmetric momentum-space structure in the form of concentric rings. The spectrum is partially symmetrized and visually enhanced by taking $\nabla^2 I$, where $I$ is photoemission intensity. The rings are also distinguishable from the raw spectrum as presented in Supplementary Fig. 3. Accordingly, the dispersive structure is only repeated along the radial direction, forming annular regions at larger momenta.

The repetition phenomenon is reminiscent of that occurring in crystalline systems, which feature duplicated dispersions commensurate with reciprocal lattice vectors, outside of the first Brillouin zone. Hence, we refer to the regions where duplicates appear as "Brillouin-like zones" (BLZ) because they demonstrate repetition akin to crystalline fermiology. Unlike for crystals, the repetitions occur only along the radial direction, which we interpret as a manifestation of the rotational symmetry expected in amorphous structures. Figure 2d conceptualizes the BLZ, showing the typical BZ-relationship of the replicated bands with respect to a reciprocal lattice constant (upper panel) and the annular zones in the Fermi surface (lower panel). The essential difference in the amorphous case is that its uniformity at long length-scales implies that reciprocal-space structure is rotational

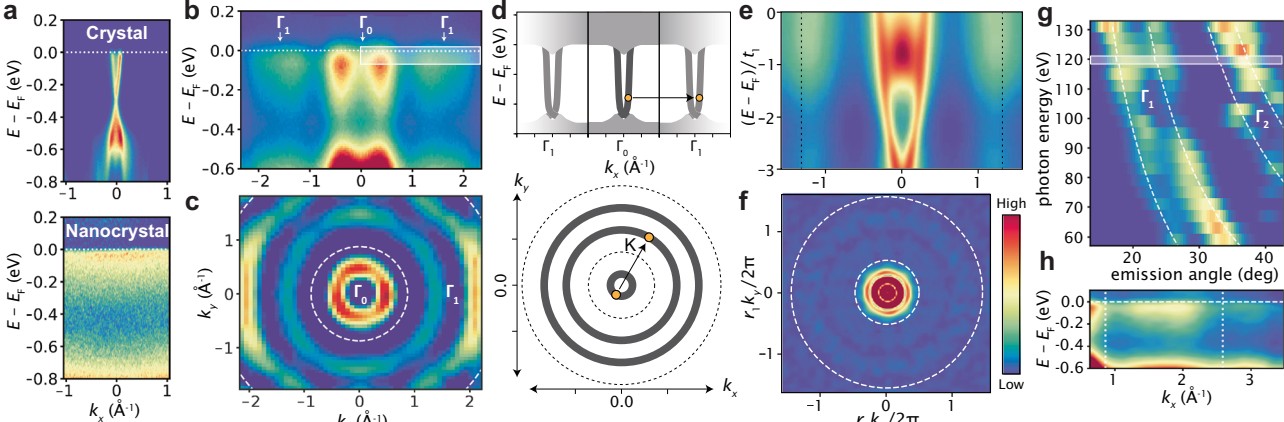

**Fig. 2 | Fermiology of the amorphous surface state. a** ARPES spectra of crystalline and nanocrystalline $Bi_2Se_3$. **b** Large momentum range ARPES spectrum of amorphous $Bi_2Se_3$ uncovers duplicate dispersions approximately 1.75 Å⁻¹ from $\Gamma$. **c** The Fermi surface ($\nabla^2 I$ of the raw intensity for visibility) demonstrates rotational symmetry of the primary and duplicated dispersion. **d** Illustration of amorphous dispersion and Brillouin zone-like repetition contingent on a characteristic momentum. **e** Simulated dispersion along $k_\parallel$ through $\Gamma$ showing duplicated structures. **f** Fermi surface from simulations showing repeated annuli. **g** Photon energy dependence of the 2nd and 3rd BZ dispersion, obeying $k_z$-independent photoemission (dashed white). **h** ARPES spectrum at $h\nu = 120$ eV with repeated dispersions separated by 1.75 Å⁻¹.

symmetric. Therefore reciprocal lattice vectors cannot exist, otherwise there would be well defined preferential directions and therefore long-range order. However preferential momentum scalars can exist, since local ordering, such as typical bond lengths, can persist in randomized directions. As a note, an effect which is absent in our spectra is the possible long-range nematic ordering. This can occur for instance via a compression or a strain which modifies the nearest-neighbor distances along one axis resulting in an elliptic transformation of the BLZ[22,28].

Expanding further out in momentum space exposes higher order BLZ. Figure 2g shows the photon energy dependence of the dispersion at the Fermi level along the same radial direction as panel (b) at large detection angles. The white dashed lines represent the photon energy dependence for 2D states at four different $k_z$-independent momenta, indicating that the features are in fact photoemission from 2D surface states as opposed artifacts from photon energy-dependent matrix elements. The first two curves from the top-left follow the 1st order BLZ (0th order being at $\Gamma_0$) and the last two curves follow the 2nd order BLZ. Panel (h) displays the momentum space converted spectrum for $h\nu = 120$ eV (shaded region in (g)) in which the bright 0th order dispersion is cutoff on the left edge and the 2nd order BLZ can be seen near 3 Å$^{-1}$. The intensity of the BLZs decrease and the dispersions broaden at larger momenta.

To determine the origin of the BLZ we compare the results of the ARPES experiment on a-Bi$_2$Se$_3$ to a numerical simulation with a tight-binding Hamiltonian of a-Bi$_2$Se$_3$ introduced in[25]. However, to explain our ARPES observations we need to ensure that a degree of local order is preserved when defining the atomic arrangement. To do so we construct a 3D arrangement of amorphous sites using thermalized hard packed spheres as in ref. [25] to which we add a relaxation step, resulting in a radial distribution function with peaks corresponding to nearest, second-nearest, and third-nearest neighbors. The peak locations define the characteristic length scales of the system. We use the nearest-neighbor average distance $r^*$ to set the scale of the plots in position space. This choice defines the characteristic momentum scale $k^* = \frac{2\pi}{r^*}$, which we use to normalize momentum space.

On this atomic site distribution, we define a model with a spin $\frac{1}{2}$ degree of freedom and two orbitals per site, to generalize the crystalline Bernevig-Hughes-Zhang (BHZ) model[29], For each site in the amorphous structure we find the six closest neighbors. Following[30], we then choose the coupling between neighboring sites so that, if the lattice was cubic, it would result in the original BHZ model. Because the hoppings are assigned sequentially for every site, the six-fold coordination is preserved on average. The hopping term of the Hamiltonian depends on the relative position of the two sites $\mathbf{d}_{ij}$ according to

$$\langle i|H|j \rangle = it_1(\hat{\mathbf{d}}_{ij} \cdot \boldsymbol{\sigma})\tau_x - t_2\sigma_0\tau_z, \tag{1}$$

while the onsite energy of a single site reads

$$\langle i|H|i \rangle = M\left(\sigma_0\tau_z + \alpha e^{-\frac{\delta_i}{2r^*}}\sigma_0\tau_0\right). \tag{2}$$

This system shows a topologically non-trivial gap that hosts a Dirac cone surface state for $M$ positive and close enough to zero[25]. The persistence of the topological state is expected due to its robustness to disorder[5,30–32]. The parameter $\alpha$ controls the strength of a symmetry-allowed surface on-site potential that shifts the surface Dirac cone away from $E = 0$, where $\delta_i$ is the distance from site $i$ to the surface.

In order to study the momentum-space structure of the electronic states, we compute the spectral function of this system by projecting into a basis of plane waves of light momentum $k$, illuminating a single surface with a finite penetration depth, simulating an ARPES experiment. One can then define the two parallel and the perpendicular components of the momentum without ambiguity. Due to the isotropy

of amorphous systems, the problem is invariant up to an in-plane rotation. One can thus only focus on the incident plane of light. The ARPES numerical simulations are computed for numerical samples containing 1200 sites, and averaged over 100 disorder realizations. The spectral function is obtained using the kernel polynomial method[33], using 200 moments. Averaging over multiple disorder realization allows to take into account the self-averaging effect occurring in the experiment due to much larger samples. In Supplementary Note 7, we show that the phenomenology we discuss next also applies to bulk states in two-dimensional systems, e.g. amorphous graphene[24].

The ARPES spectra of a-Bi$_2$Se$_3$ and the numerical model (Fig. 2e) both show the bulk gap, and within it a dispersive surface states that cross the gap around the $\Gamma$ point. Around each momenta commensurate with the characteristic momenta $k^*$, $\|\mathbf{k}_\parallel\| = k^*$, $2k^*$... a copy of the surface Dirac cone appears. At energies close to the Fermi level, the bulk states also get copied, showing that repetitions occur irrespective of the topological character of the electronic states. However, topological protection does prevent the broadening of the surface band, making its dispersion clearer than that of the bulk bands. The repetitions are the amorphous equivalent of the Brillouin zones of a crystal, enabled by the characteristic nearest-neighbor distance retained by the amorphous structure. This suggests that in Fig. 2b, c, the repetitions we observe originate in the local order of the atomic sites. Figure 2f shows that the expected spectrum is isotropic in the two components of the momentum that are parallel to the illuminated surface, as also observed experimentally. This theoretical analysis combined with the experimental spectra strongly supports the conclusion that ARPES can be used as a tool to extract the scale of local order of any non-crystalline solid by observing BLZ repetitions.

It is illustrative to compare the momentum space structure of a-Bi$_2$Se$_3$ with c-Bi$_2$Se$_3$ and to separate the roles of bulk and surface states. In Fig. 3a, we show deep binding-energy ARPES spectra at $h\nu = 120$ eV for c-Bi$_2$Se$_3$ along the $\Gamma - K$ direction and for a-Bi$_2$Se$_3$ radial from $\Gamma$. The intensity of the features near the Fermi level in a-Bi$_2$Se$_3$ has been enhanced by 10 × for visibility. The most notable difference close to the Fermi level is that the Dirac state in the crystal and the vertical features in the amorphous system have markedly different Fermi wave vectors ($k_F$), 0.08 Å$^{-1}$ and 0.4 Å$^{-1}$, respectively. We identify several factors that can contribute to this difference. First, a surface potential (captured by $\alpha$ in Eq. (2)) can shift the Dirac point downwards in energy, changing $k_F$ significantly. Second, the surface state Fermi velocity is not universal, and can be strongly affected by disorder, as we will exemplify later on.

Turning back to Fig. 3a, and looking deeper in binding energy we see that the crystalline sample maintains strongly dispersive features, whereas the amorphous sample demonstrates flattened and broadened bulk band structure. Notably there is a broad nearly-flat structure near −2 eV. Even though the curvature of the deep binding energy band structure is reduced in the a-Bi$_2$Se$_3$, the angle integrated spectral response is intriguingly similar. In Fig. 3b we plot the X-ray photoemission spectroscopy (XPS) spectra for c-Bi$_2$Se$_3$(red) and a-Bi$_2$Se$_3$(blue) on shifted y-axes for visibility. The XPS signal of the valences states provides information on bonding environment. The overall intensity as a function of binding energy follows nearly identical large-scale behavior with a dip near −8 eV and a broad shoulder with substructure at −3 eV. The crystalline sample exhibits four peaked features in the shouldered region that correspond to four peaks in the amorphous spectrum shifted by ~ 0.5 eV. The similarity between the spectra indicates that the deep binding energy band structure of the amorphous sample appears as the momentum averaged band structure in the crystal, resulting from similar local environments.

The peculiar difference between the highly dispersive bands near the Fermi level and the weakly dispersive bands at high binding energy in a-Bi$_2$Se$_3$ can be explained by reflecting on the uncommon features that a rotationally symmetric system imprints in ARPES. Figure 3c

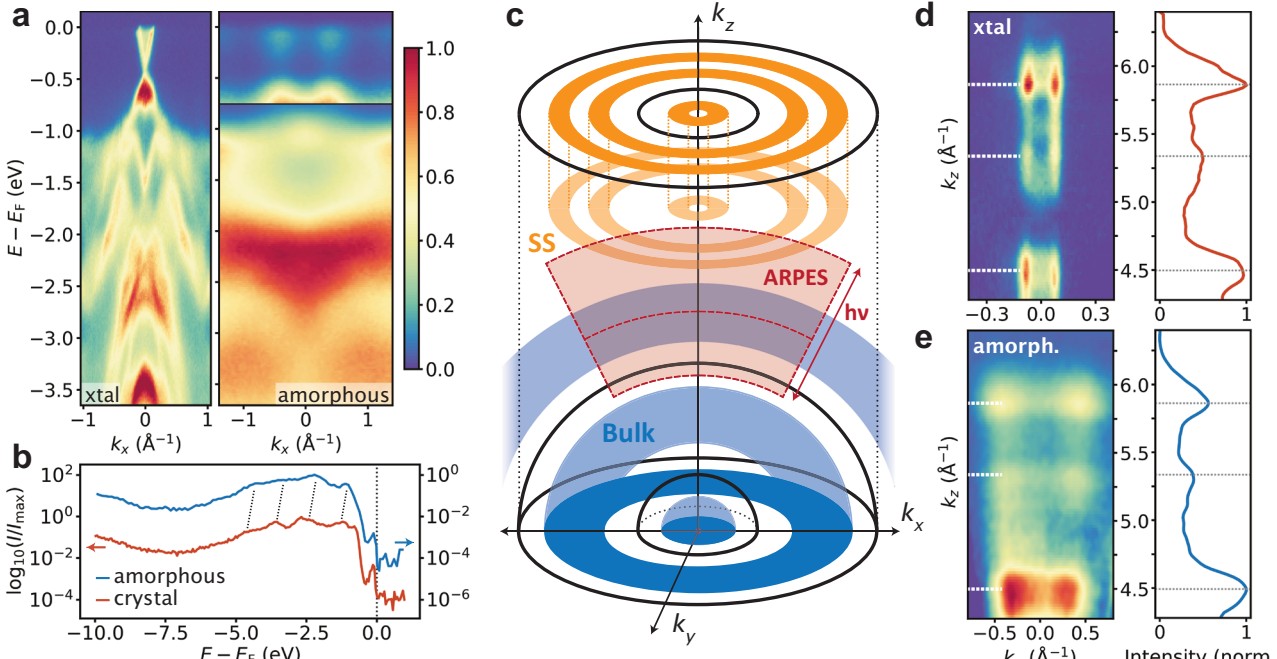

**Fig. 3 | Bulk and surface state comparisons to crystalline Bi₂Se₃. a** Deep binding energy ARPES spectra for c-Bi₂Se₃ (xtal) (Γ − K) and a-Bi₂Se₃. **b** XPS on a-Bi₂Se₃ (blue) and c-Bi₂Se₃ (red) displays similar spectra for valence bands. Dashed lines indicate corresponding peaks, and the spectral hump of the upper energy portion of the surface state can be seen near $E_F$ in both samples. **c** Diagram of amorphous band structure geometry. Bulk states form spherical shells around $\vec{k} = (0,0,0)$ (blue), whereas surface states form cylindrical shells around the $k_z$ axis (orange). For a single photon energy, ARPES probes a section of an approximately spherical shell about $\vec{k} = (0,0,0)$ (red). **d, e** Surface state spectrum at $E_F$ as a function of $k_z$ for c-Bi₂Se₃ (xtal) (**d**) and a-Bi₂Se₃ (**e**). $k_x$-integrated intensity shown to the left with 3 characteristic peaks marked by dashes.

illustrates the 3D momentum space structure of bulk bands and surface states given the symmetry of an amorphous system. The bulk bands (blue) are rotationally symmetric and are allowed to vary along the radial direction contingent on a well-defined characteristic momentum, thereby forming repeated spherical shells at constant energy within BLZs. Surface states (orange), spatially localized at the surface, are $k_z$-independent and forming cylindrical shells oriented along the $k_z$-axis. ARPES is well suited for studying $k_z$-dependence in crystals with Cartesian Brillouin zones by varying the photon energy, since:

$$k_z \propto \frac{1}{\hbar}\sqrt{2 \times m_e(h\nu - E - \phi + V_0)}. \qquad (3)$$

where $E$ is the binding energy of the electron, $V_O$ is the fixed inner potential of the material, $\phi$ is the material work function, $h\nu$ is the photon energy, and $m_e$ is the mass of the electron. However, in a spherically symmetric momentum-space, ARPES faces an additional challenge to observe $k_z$-dependent bulk states. In ARPES, $k_{x,y} \propto \sin\theta$ and $k_z \propto \cos\theta$, where $\theta$ is the angle of photoemission, such that ARPES probes an approximately spherical crossection in momentum space for a given photon energy. The solid red arcs in Fig. 3c illustrate variable ARPES crossections given fixed photon energies. Moreover, typical inner potentials, $V_O$, are of order 10 eV, limiting the minimum probable $k_z$ to approximately 2 Å⁻¹ so that the concentric structure at $k_z = 0$ is out of reach. All in all, this means that ARPES probes approximately spherical cross-sections of the spherically-invariant bulk band structure. Therefore bulk dispersive features will still disperse along $k_z$-axis by varying the photon energy (shaded red region) but features in $k_x$ and $k_y$ will appear flat.

In contrast, surface states are $k_z$ independent and the concentric structure can be accessed by ARPES at any photon energy (see red shaded region meets orange), revealing dispersive bands in $k_x$ and $k_y$.

In panels (d) and (e), we display the $k_z$-dependence of the c-Bi₂Se₃ and a-Bi₂Se₃ surface states near $E_F$, respectively. In the crystalline case, $k_F$ remains fixed for all $k_z$ at 0.08 Å⁻¹, manifesting as narrow vertical pillars in $k_z$ vs. $k_x$ with variable intensity due to photon energy dependent matrix elements.

Curiously, the amorphous surface state bands expand outward with increasing $k_z$ (Fig. 3e). This is a non-periodic dispersive feature that occurs over the full measured 2.5 inverse angstroms in $k_z$. The lack of periodicity over this range indicates that this momentum space feature cannot be due to a crystalline bulk structure since any such structure would need to repeat on smaller intervals than 2 Å ($\pi$/2.25 Å⁻¹). Moreover, this peculiarity does not affect the conclusion that these are surface states since, given the lack of long-range order, bulk states would necessarily form rotationally symmetric shells centered at $(k_x, k_y, k_z) = (0, 0, 0)$ and would appear nearly horizontal in panel (e). Therefore these states must be of surface state origin since they are coherent across many $k_z$ values.

Crucially, the momentum integrated intensity as a function of kz (panels (e) and (d), right plots) are nearly identical, with intensity peaks occurring at the same three $k_z$ values (horizontal lines). This indicates that the photon energy dependent matrix elements are comparable in the two systems, advocating for similar orbital character.

Using the BHZ model defined by Eqs. (1) and (2), we can compare the localization of the bulk and surface wavefunctions. Figure 4a shows the average wavefunction site occupation within a 2 Å slice of the amorphous cube for surface states between $E - E_D = -1.0$ and 0 eV (yellow) and bulk states between $E - E_D = -9.5$ and −4.0 eV (blue), where $E_D$ marks the center of the band gap. The in-gap surface states are localized to the system edges whereas the bulk states evenly fill the interior appearing completely delocalized. However, the singular wavefunctions tell a different story. Figure 4b shows a single wavefunction at −0.45 eV (yellow), again localized to the surface but delocalized along the 2D surfaces. A random selection of three bulk

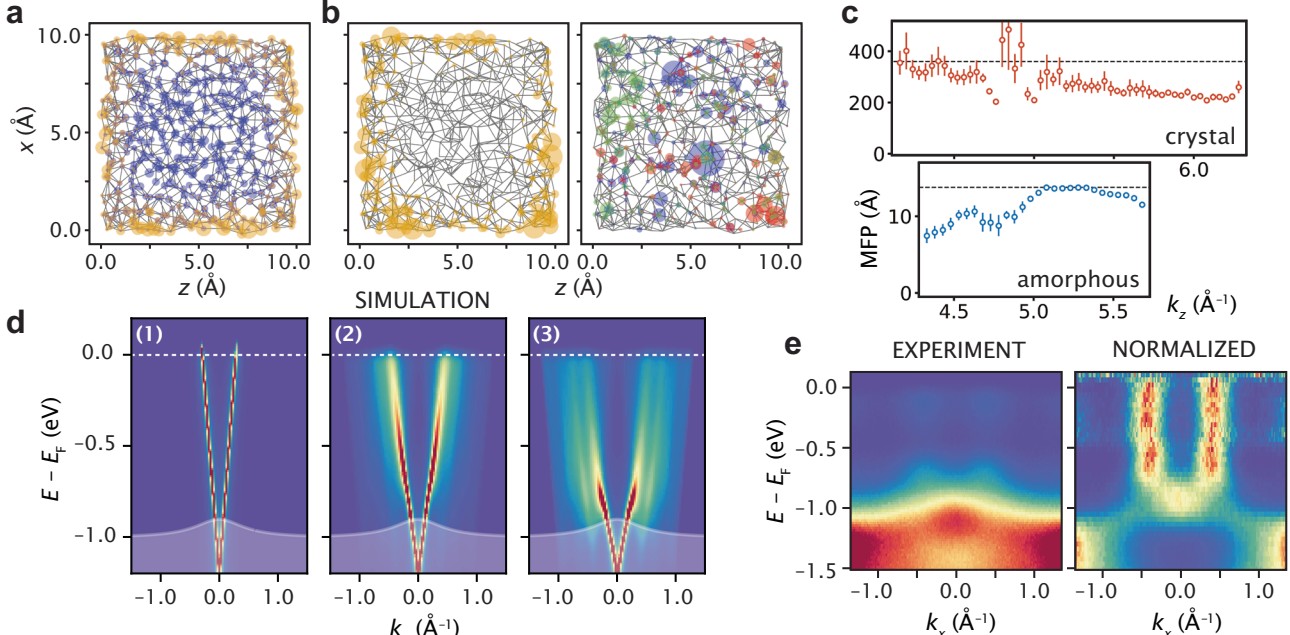

**Fig. 4 | Simulated spectral function in amorphous lattice. a** Average site occupations for wavefunctions with energy within the bulk gap (yellow) and below the bulk gap (blue). **b** Single surface state wavefunction ($E - E_F = 0$ yellow) is delocalized along the surface plane, spreading across multiple unit cells. Single bulk state wavefunctions ($E - E_F < -0.5$ eV red, green blue) are localized within the bulk. **c** Measure of mean free-path (or spatial coherence) of surface state electrons as a function of $k_z$ as determined experimentally from $2\pi/\sigma_{MDC}$ for c-Bi$_2$Se$_3$ and a-Bi$_2$Se$_3$ from Fig. 3**d**, **e**. **d** Spectral function of a linear dispersing state scattered on a

disordered array of atoms with increasing scattering strengths $v_0$. (1) no scattering (continuous medium), (2) weak atomic scattering strength, (3) full effect of the atomic lattice. When the scattering strength increases, copies of the central Dirac cone appear at the peaks of $c_2$, i.e. around $k = \pm 0.4$ Å$^{-1}$. For strong enough scattering potential, the dispersion is pushed into the valence band and broad vertical features form that cross the bulk band gap. **e** Experimental ARPES spectrum and the same spectrum normalized along the energy-axis for comparison with simulation.

wavefunctions between −9.5 and 4.0 (red, green, and blue) show a localized behavior, constrained to a small number of sites.

In fact, we are able to deduce a lower limit on the coherence length of the electronic order from the momentum broadening of the states in Fig. 3d, e[7,34,35]. The momentum Lorentzian linewidth of the two peaks in each case serve as a measure of the electronic real space ordering, in so much as perfect electronic order leads to delta functions in momentum (ignoring lifetime broadening) and spatial incoherence leads to smearing across the full BZ. This represents a lower limit for the coherence length since final state effects, spatial variations in doping, or lifetime broadening could introduce additional extrinsic broadening.

Figure 4c plots the $k_z$-dependence of the coherence length (or mean free path, MFP) as $2\pi/\Gamma_k$ where $\Gamma_k$ is the Lorentzian line-width. For c-Bi$_2$Se$_3$, MFP is largest at small $k_z$ at nearly 400 Å, serving as our estimate of the lower limit of the coherence length in the crystal. The MFP then linearly decreases towards larger $k_z$ suggesting a possible photon energy dependence of the momentum resolution. In a-Bi$_2$Se$_3$ the largest calculated MFP is near $k_z \approx 5.2$ Å$^{-1}$ which gives a value for the MFP of 13 Å. This is a markedly reduced coherence to the crystal yet coherent beyond three nearest neighbors, $3a^*$. Interestingly, the measured coherence length reduces towards lower and higher $k_z$ values indicating an additional broadening mechanism in the low momentum regime as compared to c-Bi$_2$Se$_3$.

The delocalization of the surface state along the surface planes enables the electronic wavefunction to encompass many atomic sites, which in turn enables dispersive coherent structure in momentum space. In contrast, the localization of the bulk bands, inferred from Fig. 4b would suggest a flattened band structure. This is indeed the case for the amorphous structure in Fig. 3a, in which strong dispersion occurs for the surface state near $E_F$ and bulk bands deeper in binding energy are flat. It is possible that either topological protection or spin-

momentum locking of the surface states may enhance the in-plane coherence for a-Bi$_2$Se$_3$ surface states, making it a particular good material for observing a defined amorphous band structure.

Lastly we discuss why the amorphous surface state may exhibit broad near-vertical dispersive features at a dramatically enhanced $k_F$ with respect to the crystalline spectrum (Fig. 3a). These broad vertical features, particular to a-Bi$_2$Se$_3$, are not fully explained by our tight-binding model used in Fig. 2e, f. A tantalizing additional effect neglected in the tight-binding approximation is that scattering on a disordered array of atoms can significantly alter the dispersion of a propagating state, creating the effect of a broad and vertical dispersion[14,15]. This is believed to be the case of liquid metals and surface electrons propagating within a disordered but correlated array of atoms on crystal surfaces[22]. Translated to our situation, the hypothesis is that the surface Dirac propagating state experiences scattering due to the correlated disorder intrinsic to amorphous structure. This scattering effect is not captured by the tight-binding model[14,15].

To explore this possibility, we calculate the lattice induced self-energy caused by the experimental radial density function of Fig. 1d to determine how a linearly dispersing surface state is affected by this spatial distribution of atoms. Ref. 14 calculated the band-structure due to a collection of scatterers with strength $v_0$ and arbitrary position, within the first Born approximation. Here we extend this approach to the full Born approximation, by calculating a self-consistent scattering self-energy

$$\Sigma_s(\mathbf{k}) = \frac{v_0^2}{\Omega} \sum_{s'} \int c_2(\mathbf{k} - \mathbf{k}') F_{ss'}(\mathbf{k}, \mathbf{k}') G_{\Sigma s'}(\mathbf{k}') d\mathbf{k}', \qquad (4)$$

where $F_{ss'}(\mathbf{k}, \mathbf{k}')$ is the overlap factor between the two bands (labeled by $s = \pm$) of the Dirac Hamiltonian, $c_2(\mathbf{k})$ is the Fourier transform of the

radial distribution function shown in Fig. 1d, and $G_{\Sigma s} = (E - s v_F |\mathbf{k}| - \Sigma_s + i0^+)^{-1}$ is the bare Green's Function of the surface Dirac cone in the diagonal basis. Such self-consistent Born approximation captures, through $c_2(\mathbf{k})$, the non-perturbative effect of the site displacements.

Figure 4d shows that indeed correlated structural disorder reshapes the linear dispersion as we increase the scattering strength $v_0$. Without scattering, the spectral function only shows a Dirac cone centered around $\Gamma$. When the scattering increases, copies of the central Dirac cone appear at the peaks of $c_2$, i.e. around $k = \pm 0.4$ Å$^{-1}$. For strong enough scattering potential, the dispersion bends into the valence (shaded white region) and vertical features form, crossing the bulk band gap. Thus the exact shape of the dispersion is strongly affected by the atomic lattice, especially in the amorphous case. In Supplementary Note 9, we compare these results with the crystalline case, further evidencing that electrons scattering with disordered atomic centers can explain the differences between our measured amorphous and crystalline ARPES spectra, shown in Figs. 2 and 3.

Comparing Fig. 4d with Fig. 4e (where the ARPES spectrum near $E_F$ is shown along with the spectrum normalized along binding energy) demonstrates how both the experiment and simulation exhibit broad vertical features that cross the gap, as well as an apparent band crossing and node at the valence band edge.

## Discussion

Our data indicates that the topological surface state found and bulk band structure features found in amorphous $Bi_2Se_3$ retain their momentum space structure, notably featuring Brillouin-zone-like repetitions. Using two complementary theoretical models, we show that the features we see arise from well defined short-range length scales, hinting at the possibility of more general phenomenon in all non-crystalline solids with this property, such as other amorphous materials, quasicrystals, and liquids. Combining photoemission spectroscopy with local probes, such as scanning tunneling spectroscopy, has the potential to reveal further aspects of local electronic order in amorphous materials, notably in the presence of finite magnetic fields. The presence of highly dispersive features in a-$Bi_2Se_3$ motivate the generalization of momentum-dependent phenomena to glassy systems, including spin-momentum locking[25], momentum-based paring in superconductivity, or new avenues to engineer flat-bands in amorphous phase-change materials[36].

Based on a-$Bi_2Se_3$ specifically, another direction of further study is the origin of the appreciable monotonic $k_z$-dependence of the surface states of Fig. 3d, which clearly differs from bulk states, but is not completely $k_z$-independent. A possible cause is the lack of translational symmetry itself. ARPES is based on the notion that translational symmetry conserves the in-plane crystalline momentum following photoemission. This may no longer be the case in amorphous samples; while continuous translations can be recovered on average to explain most of our results, discrete translational symmetry is lost, which may result in more subtle $k_z$ dependencies, and may be a unique feature of non-crystalline media (see Supplementary Fig. 5 for further details).

In summary, we have revealed highly dispersive surface electronic states on the surface of amorphous $Bi_2Se_3$, that exhibit a rotationally symmetric Fermi surface with repeated Brillouin zone-like repetitions. This is a direct fingerprint of a well-defined real-space length scale from the disordered hard packing of atoms in the amorphous structure, which corresponds to a well-defined reciprocal length scale. The amorphous analog to crystalline $Bi_2Se_3$ preserves the angle-integrated XPS spectral features yet exhibits remarkably different valence state behavior, manifesting as vertical-like features with large Fermi wavevectors. Since the presence of local chemical order is ubiquitous in solids, our work calls for a retrospective investigation of amorphous solids in search for dispersive features that reveal novel quantum effects in momentum space, previously reserved for crystals and quasicrystals alone.

## Methods

ARPES spectra were acquired from the MAESTRO $\mu$ARPES endstation (BL 7.0.2.1) and the MERLIN ARPES endstation (BL 4.0.3) at the Advanced Light Source at Lawrence Berkeley National Laboratory with photon energies between 60 and 140 eV and at temperatures below 80 K. Samples for ARPES were capped with 50 nm of selenium immediately following growth and decapped in $10^{-11}$ Torr base pressure around 120 C (where Se has a significant vapor pressure) for 15 min directly before measurement. At 120 C, we observed 5 min of observable Fresnel color changes of the surface and waited 10 min after color changing stopped to ensure there is no residual selenium. We confirmed that there is not residual selenium through angle dependent XPS (Supplementary Fig. 2). Furthermore, the decap procedure does not cause crystallization of the films[25].

## Data availability

All data are made available upon request to the corresponding author.

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

## Acknowledgements

We are grateful to S. Ciuchi, S. Franca, S. Fratini, D. Mayou, D. Muñoz-Segovia and S. Tchoumakov for discussions. This work was primarily supported by the Director, Office of Science, Office of Basic Energy Sciences, Materials Sciences and Engineering Division, of the U.S. Department of Energy, under Contract No. DE-AC02-05CH11231, as part of the Ultrafast Materials Science Program (KC2203) with secondary contributions from the Nonequilibrium Magnetism Program (KC2204) and the Electronic Materials Program (EMAT). TEM work at the Molecular Foundry was supported by the Office of Science, Office of Basic Energy Sciences, of the U.S. Department of Energy under Contract No. DE-AC02-05CH11231. A.L. acknowledges support from the Godon and Betty Moore Foundation EPiQS Initiative through Grant No. GBMF4859 for the implementation of the experimental setup. A.G.G. and Q.M. acknowledge financial support from the European Union Horizon 2020 research and innovation program under grant agreement No. 829044 (SCHINES). A.G.G. is also supported by the European Research Council (ERC) Consolidator grant under grant agreement No. 101042707 (TOPO-MORPH). D.V. was supported by the National Research, Development and Innovation Office of Hungary under OTKA grant no. FK 146499, and the Deutsche Forschungsgemeinschaft (DFG, German Research Foundation) under Germany's Excellence Strategy through the Würzburg-Dresden Cluster of Excellence on Complexity and Topology in Quantum Matter – ct.qmat (EXC 2147, project-id 392019).

## Author contributions

S.T.C. performed the ARPES measurements and the analysis. A.L. and S.C. developed the experimental premise and infrastructure. Q.M. and A.G.G. devised the theoretical modeling. Q. M. carried out the numerical and analytical calculations supervised by D.V. and A.G.G. The initial idea for this experiment came from S.T.C., P.C., F.H. and A.L. P.C. grew and characterized the films (XPS, Raman) supervised by and in consultation with F.H. E.K. performed and analyzed the HRTEM, supervised by and in consultation with M.S. All authors contributed to writing the manuscript.

## Competing interests

The authors declare no competing interests.
