## [Peer Review File · Nature Communications]

Establishing Coherent Momentum-Space Electronic States in Locally Ordered MaterialsREVIEWER COMMENTS

Reviewer #1 (Remarks to the Author):

This is a very interesting work for experts performing photoemission experiments. For a model system, the authors correlate real-space periodicity with the dispersive behavior of surface states, going from a crystal to an "amorphous" state. As pointed out by the authors, "amorphous" materials may still pertain short-range order, expressed by well-developed X-ray scattering patterns (rings in the present case) that allow extracting typical binding distances. Given that the sample yields a non-homogeneous scattering pattern, it is not surprising that also electronic states develop, which have a certain momentum-dependence.

It is not clear to me, however, why this report should warrant dissemination at a very high level and a very broad community. Just because "amorphous" systems have received little attention in angle resolved photoemission seems not appropriate. The fact that momentum-dependence of electronic states evolves already at very small length scales has, e.g., been explained before (for a 1D system in doi.org/10.1103/PhysRevB.56.10138). To some extent, the present analysis (with respect to the radial symmetry) may be related to that done in doi.org/10.1103/PhysRevB.71.161403, a comparison to the present work would be interesting.

On another note, the structural characterization of the samples used in the present study is not at the level that would rule out possible further reasons for the observations in photoemission. The TEM-diffraction is taken from a very small spot only; to show that the entire sample is "amorphous" X-ray diffraction (sampling a larger area/volume) would be in order, and should be further substantiated by EXAFS. Also structural information after the photoemission experiments are needed, to show that the (probably) high-intensity synchrotron beam (and associated secondary electrons) did not modify the sample structure.

Reviewer #2 (Remarks to the Author):

The paper submitted by Prof. Lanzara and colleagues reports the study of crystalline Bi_2Se_3 and amorphous Bi_2Se_3 by means of ARPES, TEM, and theoretical analysis. The subject of this study is basically the electronic structure of disordered systems, where there is no long-range order. Even in disordered systems, however, short-range order or medium-range order may exist. Then, a fundamental question that arises is whether electron coherence can develop in the absence of long-range order. This is important in that a certain degree of disorder is universal to materials used in our technologies, and the electronic structure of an intermediate regime between crystalline and fully random systems. The distinction between the long-range order, short-range order, and no short-range order is well introduced in this study with Figure 1a by three real-space point distributions (crystalline, random, and disordered hard pack). For the latter, there exists a set of rings in the Fourier transform image, which means the well-defined real-space length scale related to the nearest-neighbor distance. The presence of this well-defined length scale should affect the electronic structure of such systems. Overall, the subject of this study itself is very fundamental to those in the field of condensed matter physics and material science. In this sense, the subject of this paper seems suitable to Nature Communications.

However, what should be clearer is the novelty of this paper over the previous studies. It appears that the effect of well-defined real-space length scales to the band structure was discussed in Ref. 14. Although the experimental system is somewhat different, the physics discussed here seems quite related. It is not very clear to me what is the physics of dispersive bands shown in Figure 2b. It sounds like the similar phenomenology can be reproduced by the authors' theoretical analysis, but it is not clear more intuitively why the band structure looks like that (two blobs and vertical lines) by the effect of well-defined length scales? Instead of this point, bulk versus surface states have been heavily discussed, but it does not seem to be the most important point of this study.

Also, the authors claim that Bi_2Se_3 made in the nanocrystalline form has the reduced short-range order, but this structural difference has not been justified by HRTEM. To me the difference in the band structure shown in Fig. 2a, b is too abrupt. Can the authors control the range of orders by controlling the density of nanocrystalline domains?

Why there is no replica in Fig. 2b at the length scale corresponding to the first and third peaks in Fig. 1d?

Minor comments:

In page 6, the 4th paragraph, there is a typo, wheras > whereas.

Reviewer #3 (Remarks to the Author):

In the paper “Establishing Coherent Momentum-Space Electronic States in Locally Ordered Materials” the authors study, using various complementary experimental probes (ARPES, HRTEM), the electronic properties of a well-known topological insulator Bi_2Se_3 . It is not for me to judge the experimental aspect, but it seems to be a very thorough and exhaustive experimental work, which definitely deserves to be published in a specialized journal, dedicated to surface science.

The key question is: does the paper “represent important advances of significance to physics community”? - that would warrant a publication in the highest profile journal such as Nature Communications. Unfortunately, in my opinion the answer to this question is NO.

The authors claim that their paper builds a proof on how the electronic coherence appears in short-to-medium-range ordered amorphous Bi_2Se_3 . To this end they study crystal, amorphous and nano-crystalline samples. The study works extensively on the α - Bi_2Se_3 , while for nano-crystalline case they only provide one plot, the bottom panel in Fig.2a. In my opinion this is a missed opportunity: if one really wants to study thoroughly how the coherence disappears completely in nano-crystalline case to reappear in amorphous case, then they should study samples with crystal grains of various sizes. By developing a measure of how much the spectrum is broadened (vs the size of crystalline grain), one can then explain whether the disappearance of the coherent peak of the electronic wave is a cross-over or a proper phase transition. The theory of this phenomenon would have been also a valuable addition to the field, the non-perturbative theory, not just a perturbative expansion of self-energy. This is a kind of discovery that was advertised in the abstract and introduction, a promise that was not fulfilled. At present the authors can only claim that in α - Bi_2Se_3 the dispersive bands re-appear, but they themselves admit that this problem has

been tackled already many-decades ago in the case of electronic properties of liquids [Ref.15-16]. In particular the title of Ref.19 seems to be quite similar to the title of the present manuscript, so the theory of the amorphous phenomena is quite well established (I would also advise to check carefully all citations of that paper). It is then hard to claim the innovative aspect of the present work.

Instead of looking carefully at the problem of how the coherence disappears, the authors abandon the nano-crystalline sample and decided to dedicate the second half of the paper to surface states in α -Bi₂Se₃ (both Fig.3 and Fig.4 are dedicated to that). This is a long part of the paper, starting on page 4 (photon energy dependence proving surface character of some states) and ending on page 8. The authors emphasize a lot the distinct behaviour of surface and bulk states. Yet, in this entire discussion I cannot find any reference to the very simple fact that the surface states are topological and thus topologically protected against backscattering. This omission is actually quite shocking, since notions of topological states have been dominating solid state physics for nearly two decades now. This framework would naturally explain all properties of the observed dispersive states, however this would also imply that “Establishing Coherent Momentum-Space Electronic States” is not a universal phenomenon appearing in all “Locally Ordered Materials”, but something specific for topological insulators. In order to prove this more general statement, the authors should choose a material that is not a topological insulator. Of course, it is quite interesting that topological states survive in amorphous case, but that discovery had already been made and was reported in Ref. 5 and 6.

Overall, on the theoretical side I cannot find any profound insight into the proposed subject of establishing coherence of electronic waves. Thus I cannot recommend publication in Nat.Comm. I suggest authors re-submit their paper (obviously with a stronger emphasis put on the above mentioned explanation due to the topological protection) to a more specialized journal.

On the top of that there are several minor issues:

- the authors claim that the uniformity of rings in Fig.2c implies the lack of any long range order. However these rings are definitely not uniform: the inner one seems to have C3

symmetry while the outer one C2 or C4 symmetry. That would suggest that some fraction of LRO is nevertheless present

- the authors try to draw some parallels between Fig.4d (theory) and Fig.4e (experiment), which from the text is supposed to be straightforward, but I can hardly see it after spending a long while looking at these figures.

Response to Reviewers

Establishing Coherent Momentum-Space Electronic States in Locally Ordered Materials

December 20, 2023

1 Reviewer #1 (Remarks to the Author):

This is a very interesting work for experts performing photoemission experiments. For a model system, the authors correlate real-space periodicity with the dispersive behavior of surface states, going from a crystal to an "amorphous" state. As pointed out by the authors, "amorphous" materials may still pertain short-range order, expressed by well-developed X-ray scattering patterns (rings in the present case) that allow extracting typical binding distances. Given that the sample yields a non-homogeneous scattering pattern, it is not surprising that also electronic states develop, which have a certain momentum-dependence.

It is not clear to me, however, why this report should warrant dissemination at a very high level and a very broad community. Just because "amorphous" systems have received little attention in angle resolved photoemission seems not appropriate. The fact that momentum-dependence of electronic states evolves already at very small length scales has, e.g., been explained before (for a 1D system in doi.org/10.1103/PhysRevB.56.10138). To some extent, the present analysis (with respect to the radial symmetry) may be related to that done in doi.org/10.1103/PhysRevB.71.161403, a comparison to the present work would be interesting.

We thank the referee for considering our work very interesting for experts in photoemission. The referee raises first the valid point of why would this work be interesting for researchers outside of this community.

Perhaps we under emphasized how challenging it is to determine the structural properties of amorphous materi-

als accurately. This remains to this date, an outstanding open problem in glassy physics. To what extent the atomic structure determines the electronic properties of amorphous materials is a related problem, of key implications for technology. Decades of research have shown that a given diffraction pattern can be explained with different atomic structures. Only in 2020, the accurate determination of the microscopic atomic structure of an amorphous equivalent of graphene was published in Nature (Toh, C.-T. et al. Synthesis and properties of free-standing monolayer amorphous carbon. Nature Publishing Group 577, 199–203 (2020)) as it demonstrated that the commonly assumed, decades old Zacharisen random-network picture cannot explain the distribution of atoms in amorphous graphene. In 2021 another study in Nature found that atomic electron tomography reconstruction can be used to experimentally determine the 3D atomic positions of an amorphous solid (Yang, Y. et al. Determining the three-dimensional atomic structure of an amorphous solid. Nature 592, 60–64 (2021).). However, a direct observation of the atomic order shaping electronic properties of amorphous solids remains unreported. In this paper we put forward ARPES as a tool to observe the direct imprint of atomic order on the electronic properties of an amorphous solid. Because our findings are not specific to Bismuth selenide (see our appendix on amorphous graphene), and because amorphous solids are ubiquitous in technology (memories, DVDs, etc), we believe our work can serve as the key reference to use photoemission to directly visualize the imprint that local atomic order on electronic dispersion of amorphous solids. It is the direct access to the correlation between atomic and electronic order in non-crystalline systems that justifies, in our opinion a broad dissemination of our work.

With respect to the first article mentioned by the referee, PhysRevB.56.10138, the system is comprised of nanocrystallites with preferred orientations. This is a very different situation from an amorphous solid in which long range order is lost. Nanocrystallites can indeed form quasi-band-structures, and the cumulative effect of the preferred alignment of nanocrystallites will lead to apparent dispersive bands when integrating over many unit cells. Moreover, the local environment of the polycrystalline unit cell non-adjacent to a grain boundary will be identical to that of the single crystal unit cell, simplifying the band structure calculation. An amorphous system, in comparison, is not defined by grain boundaries lacks long range order due to small variations in each unit cell of bond length and bond angle. Therefore, the amorphous case cannot be assumed to be comprised of the accumulation of well-ordered subunits. It remains non-trivial that the randomized perturbations to bond-lengths, bond-angles, and additional contributions to the local environment can lead to coherent electronic structure and if so, to what degree of coherence is expected and of what shape the band structure may take. Such results and discussion cannot be found by studying polycrystalline materials

In the second citation referred by the referee, PhysRevB.71.161403, the system is again polycrystalline and not

amorphous graphite. The authors state that the grains are smaller than the beam-spot (100um). Polycrystalline graphite is common with grain sizes well above nanometer scale implying that long range order is present in each grain. Our work here shows that a similar radial dispersion is possible in an amorphous bulk solid in which long range order is completely quenched.

On another note, the structural characterization of the samples used in the present study is not at the level that would rule out possible further reasons for the observations in photoemission. The TEM-diffraction is taken from a very small spot only; to show that the entire sample is "amorphous" X-ray diffraction (sampling a larger area/volume) would be in order, and should be further substantiated by EXAFS. Also structural information after the photoemission experiments are needed, to show that the (probably) high-intensity synchrotron beam (and associated secondary electrons) did not modify the sample structure.

We believe that answers the questions raised here by the referee are answered in the structural data discussed in the main and supplemental information of [1]. Hence, in this answer we will refer to the main/supplemental references in [1] rather than the manuscript under consideration here. Supplementary figure 2(a) shows larger scale HRTEM with no signs of crystalline order. Supplementary figure 2(c) shows scanning nanodiffraction patterns taken on a 77 nm by 77 nm area on the sample showing that the sample is amorphous in the entire region. In all of the diffraction patterns there was no sign of crystalline order. In main figure 1(c) we performed the diffraction on many different spots across the sample showing amorphous patterns throughout. We performed XRD when the sample was capped post growth (plotted below) and after ARPES measurement when the sample had been decapped (main figure 1(e)). There are no peaks resulting from crystalline Bragg diffraction and no difference in structure. Also present, Supplementary figure 2(e), is a large area diffraction pattern taken post decap showing no signs of crystallization. We have done HRTEM(supp figure 10(a)), STEM(main figure 10(b)), and FEM(main figure 1(d)) on nanocrystalline samples and it is clear these films have crystalline order, and in contrast the films presented in this work are fully amorphous (see referee 3 response). Additionally, the TEM operates at 200-300 keV which is much higher energy than the synchrotron radiation used. We observed film crystallization after long exposure to 300 keV (> 10 min) but did not when operating at 200 keV. With typical electron doses of $10^6 e^- / \text{nm}^2 / \text{sec}$ in TEM, then $\sim 3 \text{ kW} / \text{cm}^2$ of energy is incident on the sample. In comparison, typical electron flux at the Maestro beamline is 10^{12} for 20 μm beamspot, resulting in just $\sim 8 \text{ kW} / \text{cm}^2$ at a photon energy of 200 eV.

Since high energy TEM does not crystallize the samples we safely assume, in addition to data which provides

Figure 1: Pre measurement XRD

conclusive evidence, the synchrotron measurements do not crystallize the sample. Unfortunately we do not have access to EXAFS measurements, but Raman is a measure of local bonding. We show in main figure 1(f) that the samples are amorphous in Raman but have a similar local environment to the crystal.

2 Reviewer #2 (Remarks to the Author):

The paper submitted by Prof. Lanzara and colleagues reports the study of crystalline Bi_2Se_3 and amorphous Bi_2Se_3 by means of ARPES, TEM, and theoretical analysis. The subject of this study is basically the electronic structure of disordered systems, where there is no long-range order. Even in disordered systems, however, short-range order or medium-range order may exist. Then, a fundamental question that arises is whether electron coherence can develop in the absence of long-range order. This is important in that a certain degree of disorder is universal to materials used in our technologies, and the electronic structure of an intermediate regime between crystalline and fully random systems. The distinction between the long-range order, short-range order, and no short-range order is well introduced in this study with Figure 1a by three real-space point distributions (crystalline, random, and disordered hard pack). For the latter, there exists a set of rings in the Fourier transform image, which means the well-defined real-space length scale related to the nearest-neighbor distance. The presence of this well-defined length scale should affect the electronic structure of such systems. Overall, the subject of this study itself is very fundamental to those in the field of condensed matter physics and material science. In this sense, the subject of

this paper seems suitable to Nature Communications.

However, what should be clearer is the novelty of this paper over the previous studies. It appears that the effect of well-defined real-space length scales to the band structure was discussed in Ref. 14. Although the experimental system is somewhat different, the physics discussed here seems quite related. It is not very clear to me what is the physics of dispersive bands shown in Figure 2b. It sounds like the similar phenomenology can be reproduced by the authors' theoretical analysis, but it is not clear more intuitively why the band structure looks like that (two blobs and vertical lines) by the effect of well-defined length scales? Instead of this point, bulk versus surface states have been heavily discussed, but it does not seem to be the most important point of this study.

Figure 4d and 4e presents an explanation for the distorted surface state, using a similar but albeit more involved interpretation of the theory utilized in Ref. 14. In this study we utilize the theory of liquid metals to describe renormalization of the surface state bands due to the total atomic structure of the amorphous system, not just an amorphous perturbation on an underlying crystalline lattice. Ref. 14 demonstrates how a liquid-like arrangement of dopants in a crystalline system (well-defined lengthscale) leads to renormalization of inherent crystalline bands. Our amorphous system takes this to a much further degree. We show that a crystalline matrix is not necessary for electronic dispersions and that liquid metal theory can be applied to bulk solid amorphous structures in order to predict electronic structure that is substantially different from the crystalline phase.

Regarding the experimental-theory agreement, we are happy to report that we have now upgraded figure 4d from a first-order to full, self-consistent Born approximation. The agreement is now much more transparent. This is strong indication that the mechanisms of scattering answer previous puzzles regarding the dispersion of the surface-state in an amorphous material. This analysis stands a concrete and clear difference with previous work, including Ref. 14. In the updated supplementary material we discuss how the presence of a typical length scale leads to a momentum-space scale where the Brillouin zone repetitions occur. At the points the zone-center dispersion and its repetitions mix, the dispersion relation broadens due to disorder, resulting in vertical features. We have also changed the explanation in the caption and in the main text to better describe the renormalization predicted by liquid metal theory and to emphasize the fact that this calculation does not intend to reproduce the exact same spectrum as the experiment show. This would require to know exactly the position of every single atom in the sample. Rather, it shows that the ring-shaped structure factor can significantly reshape and broaden the bands, an effect that is absent in the crystalline case.

Also, the authors claim that Bi_2Se_3 made in the nanocrystalline form has the reduced short-range order, but this

structural different has not been justified by HRTEM. To me the difference in the band structure shown in Fig. 2a, b is too abrupt. Can the authors control the range of orders by controlling the density of nanocrystalline domains?

Nanocrystalline films result from the nucleation of crystallites out of an amorphous matrix, which is not a smooth, continuous process thermodynamically. Therefore the abrupt change in dispersions should not be unexpected. Firstly, nanocrystalline films are less homogeneous than amorphous films since there are many small distinct regions of nanocrystalline order throughout the film that abruptly terminate versus an amorphous film where the local environment is similar and slowly varying throughout the film. Secondly, at the grain boundary in nanocrystalline films there is a large amount of atomic disorder, not the well coordinated, well defined local environment. This atomic-scale disorder can disrupt the well established momentum space length scale resulting in incoherent electronic structure. In main figure 1(d) and supplemental figure 10 of [1] we show HRTEM, scanning nanodiffraction and fluctuation electron microscopy of the nanocrystalline films. Additionally, below we show scanning nanodiffraction patterns of the nanocrystalline sample on a 77nm by 77nm area to show how inhomogeneous the polycrystalline samples. Comparison of the below figure with the amorphous equivalent in supplemental figure 2(c) demonstrates the homogeneity of the amorphous state.

Why there is no replica In Fig. 2b at the length scale corresponding to the first and third peaks in Fig. 1d?

The peaks in the reduced radial distribution in Fig. 1d correspond to the interatomic length scales in Bi_2Se_3 . Because Bi_2Se_3 is a compound composed of two different elements, the simplest unit cell requires a length scale that spans from one atomic species to the neighboring atom of the same species (i.e. Bi to Bi). This is represented by the second peak in the RDF. The Brillouin-like repetitions are related to the inverse of the unit cell length scale and not the first peak associated with the Bi-Se length scale.

Minor comments: In page 6, the 4th paragraph, there is a typo, wheras > whereas.

We apologize for the typo, it has been corrected.

3 Reviewer #3 (Remarks to the Author):

In the paper “Establishing Coherent Momentum-Space Electronic States in Locally Ordered Materials” the authors study, using various complementary experimental probes (ARPES, HRTEM), the electronic properties of a well-known topological insulator Bi_2Se_3 . It is not for me to judge the experimental aspect, but it seems to be a very

Figure 2: Nanocrystalline scanning nanodiffraction

thorough and exhaustive experimental work, which definitely deserves to be published in a specialized journal, dedicated to surface science.

The key question is: does the paper “represent important advances of significance to physics community”? - that would warrant a publication in the highest profile journal such as Nature Communications. Unfortunately, in my opinion the answer to this question is NO.

We thank the referee for considering our experimental work thorough and exhaustive. The referee raises first the valid point of why would this work be interesting for researchers outside of the surface science community.

Referee 1 raised a similar point, and we recollect here our main arguments. Perhaps we under-emphasized how challenging it is to determine the structural properties of amorphous materials accurately. This remains to this date, an outstanding open problem in glassy physics. To what extent the atomic structure determines the electronic properties of amorphous materials is a related problem, of key implications for technology. Decades of research have shown that a given diffraction pattern can be explained with different atomic structures. Only in 2020, the accurate determination of the microscopic atomic structure of an amorphous equivalent of graphene was published in Nature (Toh, C.-T. et al. Synthesis and properties of free-standing monolayer amorphous carbon. Nature Publishing Group 577, 199–203 (2020)) as it demonstrated that the commonly assumed, decades old Zacharisen random-network picture cannot explain the distribution of atoms in amorphous graphene. In 2021 another study in Nature found that atomic electron tomography reconstruction can be used to experimentally determine the 3D atomic positions of an amorphous solid (Yang, Y. et al. Determining the three-dimensional atomic structure of an amorphous solid. Nature 592, 60–64 (2021).). However, a direct observation of the atomic order shaping electronic properties of amorphous solids remains unreported. In this paper we put forward ARPES as a tool to observe the direct imprint of atomic order on the electronic properties of an amorphous solid. Because our findings are not specific to Bismuth selenide (see our appendix on amorphous graphene), and because amorphous solids are ubiquitous in technology (memories, DVDs, etc), we believe our work can serve as the key reference to use photoemission to directly visualize the imprint that local atomic order on electronic dispersion of amorphous solids. It is the direct access to the correlation between atomic and electronic order in non-crystalline systems that justifies, in our opinion a broad dissemination of our work.

The authors claim that their paper builds a proof on how the electronic coherence appears in short-to-medium-range ordered amorphous Bi_2Se_3 . To this end they study crystal, amorphous and nano-crystalline samples. The study works extensively on the a- Bi_2Se_3 , while for nano-crystalline case they only provide one plot, the bottom panel in Fig.2a. In my opinion this is a missed opportunity: if one really wants to study thoroughly how the coherence disappears completely in nano-crystalline case to reappear in amorphous case, then they should study samples with crystal grains of various sizes. By developing a measure of how much the spectrum is broadened (vs the size of crystalline grain), one can then explain whether the disappearance of the coherent peak of the electronic wave is a cross-over or a proper phase transition.

We thank the referee for this suggestion, which is similar to that raised by referee two. Unfortunately, growing nano-crystals of different sizes controllably is not always possible for every material. It is also a common miscon-

ception that the amorphous state can be thought as a gradual insertion of nano-crystalline domain. There is ample literature showing that the presence of grain boundaries, fundamentally changes the nature of scattering in nanocrystals compared to amorphous samples, which are in the ideal limit, homogeneous in all directions. For instance, Kuznetsov et al. find that "a significant difference in mobility between the amorphous and polycrystalline films is particularly noteworthy considering the same nominal film composition and very similar carrier concentration" and Yang and Wise show that a critical scattering length scale in nanocrystalline systems exists where coherent delocalized transport is lost (Appl. Phys. Lett. 97, 262117 (2010), Phys. Chem. C 119, 6, 3338–3347 (2015))

The experiment described in this work indeed focuses on the response of amorphous Bi_2Se_3 to ARPES. And the plot comparing the ARPES spectrum of amorphous and nano-crystalline samples is only intended to show that one can unambiguously distinguish the two structures, and make sure that the sample studied is amorphous.

However, the theoretical part of this work studies both amorphous Bi_2Se_3 (in the main text) and graphene (in the supplementary section S5). In the case of graphene, we model three samples with different ranges of order, and compare their radial density functions and spectral functions. As the local disorder increases in strength, the nearest-neighbour peak in the radial density function broadens and the radial structure in the spectral function damps faster. Broadening in position space and damping scale in momentum space are related via a Fourier-type uncertainty relation $\Delta r \Delta k \sim 2\pi$. The reference to this discussion was missing in the main text. We therefore rewrote the part of the text commenting Figure 2, to clearly point to the supplementary section S5.

The theory of this phenomenon would have been also a valuable addition to the field, the non-perturbative theory, not just a perturbative expansion of self-energy. This is a kind of discovery that was advertised in the abstract and introduction, a promise that was not fulfilled.

We thank the referee for this suggestion, since it has served as the basis to improve our theory, and match better the experiment. The point raised by Referee 3 is that we would have restricted ourselves to perturbative approaches to disorder. This point was indeed not clear in the manuscript of the main text, but our approach is in fact not perturbative in disorder. We note that to compute the electronic band structure of the amorphous models, we used two different approaches, that are complementary to one another. On one hand, we use a tight-binding model, building the electronic bands from localized atomic orbitals. In this case, the spectral function is computed from the full numerical Green's function, without any other approximation. On the other hand, the scattering calculation adopts the opposite perspective, considering quasi-free electrons weakly scattered by the random atomic lattice. While the scattering potential is weak (perturbative), the structure factor is chosen to differ

significantly from that the crystal. It is in the latter sense in which our scattering calculation is non-perturbative: the position of the atoms are far from a weak perturbation of those in the crystal. To clarify this point, we extended the appendix VIII. We compare the results obtained with simple structure factors describing a crystalline and an amorphous lattice. Those calculations are now solved within self-consistent Born approximation, instead of first Born approximation. The full-Born approximation significantly improves the agreement between experiment and theory compared to the first-order Born, as we discuss in the aforementioned appendix, the new Fig. 4, and below in this reply. We are grateful to the referee for steering us towards improving our theory.

At present the authors can only claim that in $a\text{-Bi}_2\text{Se}_3$ the dispersive bands re-appear, but they themselves admit that this problem has been tackled already many-decades ago in the case of electronic properties of liquids [Ref.15-16]. In particular the title of Ref.19 seems to be quiet similar to the title of the present manuscript, so the theory of the amorphous phenomena is quite well established (I would also advise to check carefully all citations of that paper). It is then hard to claim the innovative aspect of the present work.

As referee 3 points out, an extensive effort was dedicated to the study of the properties of liquid metals and amorphous solids decades prior. More specifically, several approximations were proposed to compute the density of electronic states, taking into account the correlations of the atomic positions due to local order. They are suitable for systems with a simple enough structure, as elemental liquid metals or random alloys for example. However, amorphous Bi_2Se_3 as a much richer structure, due to the two very different atoms that the unit cell contains. In our work, we extend the calculation proposed by Ref. 19 to the case of Dirac linear bands. With our numerical simulations we compute the expected spectral function based on the experimentally measured structure factor.

Instead of looking carefully at the problem of how the coherence disappears, the authors abandon the nano-crystalline sample and decided to dedicate the second half of the paper to surface states in $a\text{-Bi}_2\text{Se}_3$ (both Fig.3 and Fig.4 are dedicated to that). This is a long part of the paper, starting on page 4 (photon energy dependence proving surface character of some states) and ending on page 8. The authors emphasize a lot the distinct behaviour of surface and bulk states. Yet, in this entire discussion I cannot find any reference to the very simple fact that the surface states are topological and thus topologically protected against backscattering. This omission is actually quite shocking, since notions of topological states have been dominating solid state physics for nearly two decades now. This framework would naturally explain all properties of the observed dispersive states, however this would also imply that “Establishing Coherent Momentum-Space Electronic States” is not a universal phenomenon ap-

pearing in all “Locally Ordered Materials”, but something specific for topological insulators. In order to prove this more general statement, the authors should choose a material that is not a topological insulator. Of course, it is quite interesting that topological states survive in amorphous case, but that discovery had already been made and was reported in Ref. 5 and 6.

Our previous work [1] discussed the topological character of the surface state of amorphous Bi_2Se_3 yet did not have the tools to describe the unique traits of the amorphous dispersions, leaving general questions about amorphous dispersions unanswered. In this work, we describe and model the key features of the coherent band structure observed in the amorphous samples including the persistence of a radial dispersion of the spectral function due to local order and the Fermi velocity renormalization. Of course, since topological surface states are immune to disorder, they remain fully delocalized even in the amorphous sample. Hence, they are prominent candidates for the spectroscopic detection of dispersive states in amorphous matter. However, our theoretical study of amorphous graphene, along with previous analytical works, show that delocalized (hence dispersive) states should be expected in any material with local order.

With respect to the final sentence from the referee. Ref. 5 and 6 theoretically show that topological states can exist in the amorphous case, but do not provide experimental evidence of a real solid state amorphous material demonstrating topology. We have provided in this work and our previous work such evidence. Contrary to the referee’s statement, we argue that topological surface states are instead an ideal platform for studying amorphous band structure. The high conductivity and reduced back-scattering of the topological surface state ensures that the surface electrons are strongly delocalized, maximizing our chances to observe coherent electronic order and still without reliance on crystalline order. Moreover, the Dirac dispersion provides a simplistic canvas for testing theoretical models to describe the observed renormalization. Though it would be ideal to test a suite of amorphous materials, such a quest is much more difficult than it would appear. It took years to analyze the amorphous Bi_2Se_3 structure to the point in which we could be confident in its amorphous quality. And it took similar effort to develop a method to measure the sample without compromising the surface quality and structural integrity. Our hope is that this study invites future band structure investigation of amorphous materials with similar rigorous preparations and provides a starting point for developing further improvements to ground-up theoretical descriptions.

Overall, on the theoretical side I cannot find any profound insight into the proposed subject of establishing coherence of electronic waves. Thus I cannot recommend publication in Nat.Comm. I suggest authors re-submit their paper (obviously with a stronger emphasis put on the above mentioned explanation due to the topological

protection) to a more specialized journal.

On the top of that there are several minor issues:

- the authors claim that the uniformity of rings in Fig.2c implies the lack of any long range order. However these rings are definitely not uniform: the inner one seems to have C3 symmetry while the outer one C2 or C4 symmetry. That would suggest that some fraction of LRO is nevertheless present

Fig. 2c is a symmetrization and second derivative (in that order) plot of a large portion of the surface state. We have now provided the raw spectrum shown below in the supplementary. Some of the artifacts described above are due to the second derivative and copied over by the symmetrization. There are photoemission geometry matrix elements related to the handedness of the emission plane itself that produce non-uniform intensity across the Fermi surface and that distort the final processed image. It is unlikely that if C2, C3, or C4 symmetry in any weak form exists, that we could differentiate it from purely geometrical matrix elements given the broad signal. However, even in the case of a nanocrystalline or polycrystalline system, such a signal would be unlikely given the random orientation of the domains and the large beam spot. Instead, we have ruled out long range order using the structural measurements we have described in the main text.

Figure 3: Raw Fermi Surface

- the authors try to draw some parallels between Fig.4d (theory) and Fig.4e (experiment), which from the text is supposed to be straightforward, but I can hardly see it after spending a long while looking at these figures.

We thank the referee for raising that point. Following the Referee's suggestion we have now upgraded our scattering theory to a self-consistent Born approximation which uses the measured structure factor. The results, summarized in Fig. 4 are now much closer to what the experiment observes.

The tight-binding analysis, combined with the scattering theory draw a coherent picture which we believe deserves a prominent place in the literature. The tight-binding calculation shows that repetitions are expected with local order, and the scattering theory shows that severe renormalization of the Fermi velocity are to be expected for long-wavelength surface states. Our assumptions of local order are expected to be valid in a wide-class of topological and trivial materials, such as 2D amorphous materials, and hence we believe they are foundational to understand future photoemission experiments in disordered solids.

References

- [1] Corbae, P. et al. Observation of spin-momentum locked surface states in amorphous Bi_2Se_3 . Nature Materials (2023).

REVIEWER COMMENTS

Reviewer #1 (Remarks to the Author):

Given the re-worked text better explaining the relevance of the present contribution, and the further small changes, I would now be able to support publication of this manuscript in its present form.

Reviewer #2 (Remarks to the Author):

I have carefully reviewed the revised manuscript and the response letter to the referees' comments. In my view, authors have responded to all technical points appropriately. However, I would agree with Referees #1 and #3 that it is not clear why these findings warrant publication in a high-profile journal like Nature Communications. Like I said before, I still believe that the band structure of electrons in the amorphous systems is a fundamental issue in the study of condensed matter physics. Apart from the fact that in this work authors have tried to address this relatively unattended subject, what is a new insight we can gain from these findings?

The most important point in this work seems to be the observation of dispersive bands even in the absence of long-range order. However, it was already known that there can be a dispersive (free-electron-like) band in disordered systems with no long-range translational order like quasicrystals, for example, see Nature 406, 602 (2000). In this work, the replication of dispersive bands was also clearly observed. In that sense, the bold claim that the repeated Fermi surface structure is observed for the first time seems not exactly true. Also, it is not a new story that even for heavily disordered (crappy) samples one can still find some dispersive bands (although broad) in ARPES experiments. That is, it is not too surprising there remain some degree of electronic coherence even in the absence of a long-range order.

In response to the comments given by Referee #1 and #3, the authors emphasized that “a direct observation of the atomic order shaping electronic properties of amorphous solids remains unreported”. Even if it's true, does that observation give us some new insights that deserve to be published in Nature Communications? The authors should be able to clearly address this question.

Reviewer #3 (Remarks to the Author):

While I really appreciate the substantial work that authors have put in their re-submission, regretfully I had to say that my critical, fundamental issue (A) has not been addressed. There is also an issue with the theory (B).

A. The facts, beyond any doubt, are:

1. Surface states in topological insulators are special and they are particularly well protected against back-scattering. From the point of view of applications, this is probably the main reason why the field of topological insulators has flourished over the last decade. Moreover, it has been noticed that topological surface states survive also in the amorphous phase. As far as I can say this has first been postulated in [Phys. Rev. Lett. 118, 236402 (2017)], a paper that has been cited so far 125 times. Perhaps there were earlier publications along these lines, authors may verify it, but certainly they ought to cite this paper (which they do not).

2. Authors made their experiment on Bi₂Se₃ which is a paradigmatic, one of the first experimentally confirmed, topological insulators.

In their reply to me authors show that they are very much aware of all this and they are able to point out the fine details which differs their experiment from previous ones on Bi₂Se₃. Moreover, they seem to be using amorphous graphene as a model for their theoretical simulations, at least the reader can have an impression like that (more on this issue below, see B). This could make sense only because of critical role of Dirac-type states and in Bi₂Se₃ the topological states have the linear (Dirac) dispersion.

However, in the main text of the article the authors still claim that the role of topological protection is only to help “It is possible that either topological protection or spin-momentum locking of the surface states may enhance the in-plane coherence for a-Bi₂Se₃ surface states, making it a particular good material for observing a defined amorphous band structure”, while in the Conclusions they go ahead with very general statement “our work calls for a retrospective investigation of amorphous and other noncrystalline systems such as quasicrystals, in search for dispersive features that reveal novel quantum effects in momentum space, previously reserved for crystals alone.”.

In my opinion they do not have an evidence to support such a general statement.

From the theoretical point of view their statement would require the following: there exist some surface state (aSS) which is present in all amorphous materials (topological or not). This state is dispersive (I suppose it can have Dirac or non-Dirac dispersion). In Bi₂Se₃ this state couples with topological surface states (aTSS) and they help it to improve its dispersion (and visibility?). Or this aSS turns into topologically protected aTSS when the material is topological? And if it is this second scenario then the change aSS→aTSS is a cross-over or phase transition?

If the authors wish to have experimental support of their general aSS hypothesis then they should make the same experiment on a material that is not a topological insulator (I understand carbon-based amorphous material would be their choice). Only then, if they see the same effect, they can say that here there exist the aSS not obscured by the aTSS.

At the very least using theoretical methods they could study close structural analogues of Bi₂Se₃ that are not topological, the Bi₂S₃ is a nice case which undergoes the topological transition at a finite pressure which could in principle allow to study how the aSS is gradually modified by the presence of aTSS. Then they could see if their aSS are indeed there also in the non-topological case, or not. By the way, for the simulations of disordered samples reported in Fig.4a, I presume it is a result of numerous simulations with different atomic structure taken each time (it ought to be, one cannot make all the statement in the text based on only one “numerical sample”), but I cannot find anywhere information how many samples were taken and what is standard deviation of various extracted quantities, e.g. the mean free path, MFP.

In my opinion there exist only the α TSS and there is no reason to invoke anything else. The states that they report do not exist in topologically trivial material where all back-scattering channels are allowed.

At present the authors have in their hands a good experimental proof of dispersive surface states in topological insulator. A paper based on such a basic, simple fact is something that I could support for publication. Gladly support, taking into account the amount of experimental work invested here. Of course that would require changing title, abstract, intro and conclusions (basically entire packaging of the experimental data). It would also require a serious conversation with Nature Comm. Editors, whether or not they see such an article in their journal. Of course, on the theory level the manuscript is quite incremental (see e.g. Remark 2A); it would be for instance desirable if the theory can explain the “mysterious” k_z dependence in Fig.3e; but perhaps there is some substantial innovation on the experimental side that would nevertheless warrant the publication in Nat.Comm.

It could be perhaps acceptable to put forward the hypothesis of general α SS in a Discussion section, if authors wish to do so. For instance something along the lines: “Here we show that well defined α TSS exist in TI, but perhaps they are more general phenomenon, which is exciting avenue for further research”

Actually, at present the Discussion section is actually quite puzzling, its second paragraph (about Fig.3d-e) gives impression that k_z momentum is not a good quantum number (because translational symmetry is lost) which agrees well with basic solid-state-textbook intuition, but is against what authors claim about BLZ for the in-plane momenta. Then a cautious reader asks himself a question: if an in-plane momentum supports BLZ and the perpendicular k_z host “subtle effects” (mixing of various k_z ??) then is the amorphous phase really spherically invariant, as it is emphasized in Fig.2?

Remark1A: Please note that Supp. Fig.4 in authors own manuscript shows a case of amorphous solid without SS, namely Se-rich sample. In my hypothesis, of the α TSS only, the large amount of Se (element with smaller spin-orbit coupling) suppress the gap inversion and thus the topological protection against back-scattering.

Remark2A: the angle integrated ARPES is a local probe (I deduce it from Fourier transform's definition) so it is not at all surprising that averaged core levels match each other. It is not a discovery, rather a necessary cross-check.

B. In their reply authors made an attempt to go beyond the theory from early 1960' and computed full Born approximation, fBA, which was indeed rewarding for them as they were able to generate Fig. 4d. I am happy to inspire something that made authors exited enough to call it "tantalizing". However in doing that the authors put on the center stage their amorphous carbon calculations. In my previous review I was rather lax about this sudden, surprising appearance of carbon allotrope in the manuscript where no experiment on carbon is reported. Now I see that this was my mistake.

I think authors should seriously re-consider removing those theoretical results on carbon. First and foremost: if one makes experiment on Bi₂Se₂ then he makes ALL computer simulations on Bi₂Se₃. There must be no ambiguity here, there is no circumstance/exception/explanation that would justify comparing any experimental finding in Bi₂Se₃ with numerical results on amorphous carbon (or mono-atomic lattice as the carbon is sometimes called). Reading present text I was sometimes puzzled if the experimental plot is not compared with carbon numerics, so let me re-iterate: this must not be done. (see Remark1B below)

The logic of authors seem to be: look we have the same aSS states also in an amorphous carbon. So this must be the generic effect. However, the authors do not have a full many-body solution for neither material. Tight-binding is one of the crudest approximation in solid state physics, it is even poorer than single-particle mean-field, like DFT. The most beautiful aspect of TSS is that the topological protection switches-off several relevant electron-impurity and electron-electron back-scattering terms -- in Bi₂Se₃ this ultimately leads to the observed aTSS. In the carbon allotrope one can obtain surface state on a single particle level, but there is no reason that it will survive in a real material with all interactions included, only experiment can prove it. The authors performed the full Born approximation (fBA) calculation which is OK (actually nearly fBA because there is a strange ad-hoc alpha coefficient in the denominator in Eq. 4 in SM). But from Wilson-type RG work on Anderson localization we know that in 2D the localization appears only in RG (it is actually marginal term), to get it one thus need a parquet re-summation in perturbative language. The added-

by-hand coefficient α , which appears where the vertex correction would have been, is not that innocent at all.

To conclude: if authors would like to have evidence of the general aSS without experiment, but only based on the theory then this must be a theory with exact solution of a full many body problem. Only then there would be no loophole in the reasoning. This is probably even more difficult than the experimental evidence.

In here the authors do not even study how the topological protection modifies their fBA. Hence their argument is massively insufficient.

Remark1B: The fact that two materials host Dirac states does not make them equal, not even in a reciprocal space. In Bi_2Se_3 the Dirac states are due to spin-orbit interaction and are limited to in-gap energy range while in graphene they are in a much broader energy range and due to an underlying hexagonal crystal structure (and there is no spin-orbit coupling on an element as light as carbon). Even if we only build an effective theory in reciprocal space, in Bi_2Se_3 one has to take carefully into account the spin conservation in scattering which severely limits scattering phase space. One could continue long with such a list of differences...

Here it is even worse: the authors pertain to study the role of variation of atomic positions. And on this microscopic level Bi_2Se_3 is completely different than monoatomic carbon! The analogy (if any) completely breaks down.

Second Response to Reviewers

Establishing Coherent Momentum-Space Electronic States in Locally Ordered Materials

May 8, 2024

1 Reviewer #1 (Remarks to the Author):

Given the re-worked text better explaining the relevance of the present contribution, and the further small changes, I would now be able to support publication of this manuscript in its present form.

We are grateful to the referee for the constructive dialogue and for recommending publication.

2 Reviewer #2 (Remarks to the Author):

I have carefully reviewed the revised manuscript and the response letter to the referees' comments. In my view, authors have responded to all technical points appropriately. However, I would agree with Referees #1 and #3 that it is not clear why these findings warrant publication in a high-profile journal like Nature Communications. Like I said before, I still believe that the band structure of electrons in the amorphous systems is a fundamental issue in the study of condensed matter physics. Apart from the fact that in this work authors have tried to address this relatively unattended subject, what is a new insight we can gain from these findings?

We are glad that the referee considers our technical answers satisfactory. As the referee points out the band structure of electrons in the amorphous systems is a fundamental issue in the study of condensed matter physics. One could phrase the fundamentally broad question we experimentally tackle as: how different is the electronic

dispersion in crystals, quasicrystals and amorphous? Our direct observation of the existence of quasi-Brillouin zone like repetitions in the electronic dispersion confirms that this phenomenon is not unique to crystals or quasicrystals. We provide the first direct link between electronic and short-range structural order in amorphous solids. We must remember that this is still a point of contention in the literature. As in the quasicrystalline paper cited by the referee below, a typical conclusion in the literature seems to be that amorphous systems have necessarily localized all electronic states, and hence their electronic structure is non-dispersive and uninteresting compared to crystals and quasicrystals. Without experimental evidence, the community has forgotten that short-range order can lead to Brillouin-zone repetitions. This was already discussed first by Edwards in the sixties [1, 2]) and later theoretically rediscovered by others [3]. Our theory significantly expands these findings (see answer to Referee 3). Our paper fills the lack of necessary experimental evidence and finally confirms that reciprocal-space spectral repetitions are an ubiquitous phenomenon in all short-range ordered systems. Without our work only quasicrystalline and crystalline systems have been confirmed to display this phenomenon. Because such a fundamental phenomenon is still taught as a crystalline phenomenon in solid-state text-books, we believe that our findings should appeal to any condensed matter physicists, and thus deserves broad dissemination.

The most important point in this work seems to be the observation of dispersive bands even in the absence of long-range order. However, it was already known that there can be a dispersive (free-electron-like) band in disordered systems with no long-range translational order like quasicrystals, for example, see Nature 406, 602 (2000). In this work, the replication of dispersive bands was also clearly observed. In that sense, the bold claim that the repeated Fermi surface structure is observed for the first time seems not exactly true. Also, it is not a new story that even for heavily disordered (crappy) samples one can still find some dispersive bands (although broad) in ARPES experiments. That is, it is not too surprising there remain some degree of electronic coherence even in the absence of a long-range order.

We thank the referee for this remark and bringing this reference to our attention. Incidentally, we point out that the novelty of this reference hinges in contrasting the dispersing quasicrystalline spectral properties with "localized", non-repeating spectral properties in amorphous solids. These statements, apparent in their abstract and throughout the text, serve as a perfect example of what we alluded to in our previous reply. The community, as exemplified in that paper specifically, seems to keep resurfacing the idea that amorphous spectral properties do not repeat, and hence are uninteresting. We directly observe that this is not the case.

It is also important to remark that quasicrystals do have long-range order: the Fourier transform of their lattice, the structure factor, has well defined peaks. Sharp peaks are a direct manifestation of their long-range order. We do agree that they have quasiperiodic translational order, and hence these peaks are quasiperiodic, and not periodic. Quasicrystals also display, although not always, rotational symmetries not compatible with long-range crystalline translations, as observed in the cited reference.

In contrast, amorphous systems truly lack long-range order even when defects are minimal. Zachariasen described in his famous paper from 1932 [4] that glasses can be defect free from the perspective of local bond coordination, yet exhibit only short range order. Their structure factor is ring-like, signifying rotational invariance, absence of long-range order, but characteristic short-range order. We of course agree with the referee that disordered samples can show broad bands. However, we have argued above that the fact that amorphous systems still display disperse bands and repetitions is not widely accepted, as can be read in the cited reference. We attribute this to the lack of experimental evidence to support this idea. Prior to our work, there was no experimental answer to whether aspects of Bloch's theorem persist in the absence of periodic and quasi-periodic order. Edwards' theory hinted at this possibility, but a direct link between electronic and structural order in amorphous solids was lacking. This is what our work offers, a direct observation of this phenomenon.

In the spirit of the constructive remarks of the referee, we have toned down the claim in the abstract that the repeated Fermi surface structure is observed for the first time and particularized to amorphous matter. Since amorphous solids are a significant fraction of all known solids, and we believe this phenomenon is generic (see reply to Referee #3) and we believe it warrants broad dissemination.

Changes in the main text: We have toned down the abstract by better specifying that our findings pertain to amorphous solids: *Moreover, we observe for the first time in an amorphous material[...]*

In response to the comments given by Referee #1 and #3, the authors emphasized that “a direct observation of the atomic order shaping electronic properties of amorphous solids remains unreported”. Even if it's true, does that observation give us some new insights that deserve to be published in Nature Communications? The authors should be able to clearly address this question.

We hope that with the above answers we have addressed this question. In short, the existence of Brillouin-zone repetitions of dispersive bands have been often portrayed as a consequence of long-range order, i.e. to crystals and quasicrystals. However, delocalized, dispersing states can exist in real samples also in amorphous solids, that

lack long-range order. Our work is the missing experimental evidence that this is indeed the case. It completes the link between electronic band-structure repetitions and short-range structural order in all three types of solids: crystalline, quasicrystalline and amorphous solids.

3 Reviewer #3 (Remarks to the Author):

While I really appreciate the substantial work that authors have put in their re-submission, regrettably I had to say that my critical, fundamental issue (A) has not been addressed. There is also an issue with the theory (B).

We appreciate the substantial work that the referee has put in their understanding of our paper. While we have disagreements with some points brought up by the referee, we try to bridge the misunderstanding by clarifying further our points below. We concede that these might not have been presented clear enough, especially the role played by the graphene example.

We are glad to read that the referee considered that *"At present the authors have in their hands a good experimental proof of dispersive surface states in topological insulator. A paper based on such a basic, simple fact is something that I could support for publication. Gladly support, taking into account the amount of experimental work invested here."*

To be clear: We claim that whenever short-range order in real space exists, there is an associated momentum scale. This causes Brillouin-zone-like repetitions. This is the unexpected finding we report in this work. One could rather expect that the lack of long-range order results in a very broad bands, with no momentum dependence. We believe the prominent role of local order was not reported experimentally before for amorphous systems. Moreover, preparing samples that are truly amorphous, characterized as amorphous through TEM, Raman, x-ray diffraction, etc, and with a suitable surface for ARPES given the same conditions as the structural characterizations required is a feat that took years of preparation. Thus, our experiments visualize these repetitions directly and our theory aims to show that this is a generic phenomenon by considering two examples, amorphous bismuth selenide and graphene. We hope our work motivates additional groups to further validate the claims in more material systems.

Changes in the text: We have added a section at the beginning of the supplemental materials clarifying the novelty of our work and how each measurement and calculation link together.

A. The facts, beyond any doubt, are: 1. Surface states in topological insulators are special and they are particularly well protected against back-scattering. From the point of view of applications, this is probably the

main reason why the field of topological insulators has flourished over the last decade. Moreover, it has been noticed that topological surface states survive also in the amorphous phase. As far as I can say this has first been postulated in [Phys. Rev. Lett. 118, 236402 (2017)], a paper that has been cited so far 125 times. Perhaps there were earlier publications along these lines, authors may verify it, but certainly they ought to cite this paper (which they do not).

We thank the referee for this remark. We are indeed aware of this paper and should have cited this work and we have included it in the updated version. We would like to point out that this paper chooses to study a random lattice. This is not a realistic model for amorphous solids, a matter which is at the core of our paper. Amorphous solids retain short-range order due to the chemical bonding of the atoms. Other works have considered amorphous models with tightly packed sites and well defined bond-lengths [5, 6] ([5] was already cited as pioneering as the one mentioned by the referee). We discussed the differences between different models in our recent review Ref. [7] which we now cite in the new version of the manuscript. While previous works demonstrate theoretically that topology persists in randomized lattices, these works overlooked the necessary link between the short-range real-space order and momentum space coherence, which is key to our work.

Changes in the main text: We have added references [6, 7, 8] to the main text. In our discussion of Fig. 2, we now emphasize the fact that the robustness of the topological surface state is advantageous, but that it is not necessary for Brillouin-zone-like repetitions to occur.

2. Authors made their experiment on Bi_2Se_3 which is a paradigmatic, one of the first experimentally confirmed, topological insulators.

In their reply to me authors show that they are very much aware of all this and they are able to point out the fine details which differs their experiment from previous ones on Bi_2Se_3 . Moreover, they seem to be using amorphous graphene as a model for their theoretical simulations, at least the reader can have an impression like that (more on this issue below, see B). This could make sense only because of critical role of Dirac-type states and in Bi_2Se_3 the topological states have the linear (Dirac) dispersion.

We use graphene as an example to show that quasi Brillouin zone repetitions can be observed whenever there is local order, independent of topology. We have to emphasize that it has nothing to do with its Dirac character, and has been chosen because of its experimental relevance (see e.g. [9]). There is nothing special about graphene and Bi_2Se_3 in the sense that any dispersion relation upon scattering with correlated disorder will have Brillouin zone

repetitions, not matter their protection or topological properties. This is explicit in the main text (Fig. 2.e) and in the appendix (e.g. Figure S4). In particular, in Fig S4 1,o, we show that the repetitions occur for energies far away from the Dirac cones, e.g. for states at the top of the band. These repetitions are what we observe directly here for an amorphous solid for the first time, for the particular example of topological surface states in Bi_2Se_3 .

Changes in the main text: We have rephrased the explanation of Fig. 2e to point out that both the topological surface state and the surrounding bulk states get repeated at momenta commensurate with k^* .

However, in the main text of the article the authors still claim that the role of topological protection is only to help “It is possible that either topological protection or spin-momentum locking of the surface states may enhance the in-plane coherence for a- Bi_2Se_3 surface states, making it a particular good material for observing a defined amorphous band structure”, while in the Conclusions they go ahead with very general statement “our work calls for a retrospective investigation of amorphous and other noncrystalline systems such as quasicrystals, in search for dispersive features that reveal novel quantum effects in momentum space, previously reserved for crystals alone.”.

In my opinion they do not have an evidence to support such a general statement.

From the theoretical point of view their statement would require the following: there exist some surface state (aSS) which is present in all amorphous materials (topological or not). This state is dispersive (I suppose it can have Dirac or non-Dirac dispersion). In Bi_2Se_3 this state couples with topological surface states (aTSS) and they help it to improve its dispersion (and visibility?). Or this aSS turns into topologically protected aTSS when the material is topological? And if it is this second scenario then the change $\text{aSS} \rightarrow \text{aTSS}$ is a cross-over or phase transition?

We thank the referee for highlighting this point. First it is important to note that both surface and bulk states, e.g. the conduction band, present Brillouin-zone-like repetitions. This is seen in Fig. 2b, d for Bismuth selenide and in Fig. S4 for the (bulk) band-structure of graphene. Therefore, it is not a surface phenomenon. There is no need to invoke trivial surface states coupling to topological ones as the referee seems to imply in their comment. To be precise, our statement is that given short or mid-range order, one should expect Brillouin repetitions, for *both* bulk and surface states. We believe we have good theoretical support for this general statement, given our results in graphene and Bismuth selenide models. As happens with any phenomenon, there are materials more advantageous to observe it than others. We chose amorphous Bi_2Se_3 as its topological properties will only make the phenomenon

we discuss more likely to be experimentally observable due to the reduced phase space for scattering.

Lastly, we note that ARPES is better suited to reveal the 2D cylindrical momentum-space symmetry of surface states more clearly than the bulk 3D spherical symmetry (see Fig. 3c). This may be one reason why such coherence has not been previously reported in glassy band structures.

If the authors wish to have experimental support of their general aSS hypothesis then they should make the same experiment on a material that is not a topological insulator (I understand carbon-based amorphous material would be their choice). Only then, if they see the same effect, they can say that here there exist the aSS not obscured by the aTSS. At the very least using theoretical methods they could study close structural analogues of Bi₂Se₃ that are not topological, the Bi₂S₃ is a nice case which undergoes the topological transition at a finite pressure which could in principle allow to study how the aSS is gradually modified by the presence of aTSS. Then they could see if their aSS are indeed there also in the non-topological case, or not.

First, we do not claim, also do not expect that, all amorphous materials have surface states. We do claim that whenever short-range order in real space exists, there is an associated momentum scale. This causes Brillouin-zone-like repetitions that we observe, in this case at the surface of amorphous Bi₂Se₃. Our theory in graphene supports that the repetitions are generic.

We used graphene as a pedagogical example. It is purposely far from Bi₂Se₃: it is 2D, it is a metal and it is not topological. Real amorphous graphene has short range order (see the experimental realization in e.g. [9]), and we predict it should show similar Brillouin-zone-like repetitions. If the referee considers it irrelevant we can remove it, but we believe it makes the point that short range order leads to repetition in a simple way, without the need to invoke topological surface states.

By the way, for the simulations of disordered samples reported in Fig.4a, I presume it is a result of numerous simulations with different atomic structure taken each time (it ought to be, one cannot make all the statement in the text based on only one “numerical sample”), but I cannot find anywhere information how many samples were taken and what is standard deviation of various extracted quantities, e.g. the mean free path, MFP.

First we would like to clarify that the main points of our paper do not depend on averaging over disorder realizations. The existence of Brillouin zone repetitions hinges on the existence of a well-defined real-space scale and thus should appear in any disorder realization, provided the sample is big enough. In other words, the existence

Figure 1: Average localization of gap (with energy $-2.1 < (E - E_F)/t_1 < -1.8$) and band states (with energy $-8 < (E - E_F)/t_1 < -7.5$). The color shows the relative weight of each $\langle \psi_{gap}|i\rangle^2 / (\langle \psi_{gap}|i\rangle^2 + \langle \psi_{band}|i\rangle^2)$, on a given site $|i\rangle$. The site positions of 100 different disorder realizations are superimposed on the same plot.

of Brillouin zone repetitions is a self-averaging property. To answer the referee, we here compare the results between single realizations and disordered averaging.

In what concerns Figure 4, we aim to show that there are typical surface states localized to the surface that live in an energy range within the bulk gap. Our simulations of disordered samples indeed generate randomly-chosen and different configurations for each run. As indicated in the caption, Figure 4a and b show the typical localization of the eigenstates for one sample realization. This lattice realization is drawn in the background of Figure 4a and b. Figure 4a is averaged over two energy ranges, one in the gap (in yellow), one in the band (in purple). To answer the referee's question, we show in the figure 1 of this reply the average localization of states for 100 disorder realizations. This plot shows that gap states remain at the surface independent of the disorder realization. We point out that due to the amorphousness, different atomic sites in a single sample actually show different local disorder realizations. We are willing to change figure 4a with the version showed here, if the referee considers it appropriate. However, we think that averaging over multiple configurations does not help the readability of the figure, but rather makes it harder to understand. In this case, showing a single realization does not jeopardize the generality of the claim regarding the fact that in-gap states are surface states.

This interesting point raised by the referee also applies to the tight-binding plots in Figure 2e and 2f. In the previous version of the manuscript, the plots we presented aimed to describe as precisely as possible the band structure of our numerical sample. Thus, we showed the spectral function of rather large samples, with about

3500 atomic sites, computed with the kernel polynomial method [10] using 600 polynomial moments. This grants an energy resolution of about $1.5 \times 10^{-2}t_1$. Thus, in Figure 2e, we were able to distinguish multiple copies of the central Dirac cone coming from the different local configurations coexisting in the sample, grouped around commensurate momenta with $\frac{2\pi}{r_1}$. Hence, Figure 2e and 2f showed both the global repeating structure we observe in the experiment, which does not depend on realization, and also finer details that actually depend on the disorder realization. This might be the origin of the confusion, and we apologize for not being clear about it. We were interested on the broad repetitive structure and not on the realization-dependent fine structure.

In order to clarify this point in a constructive spirit, we propose to change the plots showing the spectral function of our tight-binding simulation to disordered averaged ones. We have decreased the resolution, using only 200 polynomial moments, and we have now averaged over 100 disorder realization in a smaller sample (1200 atomic sites). Averaging over multiple realizations allows us to take into account the self-averaging effect occurring in the experiment. Note that, as expected, Brillouin zone repetitions still occur, and appear independent of the disorder realization (see Fig. 2 in this reply). The disordered average results make Figure 2e and 2f in the main text resemble even more closely the experiment and hence we have incorporated them in the new version of the manuscript.

Changes in the text: We redrew all the ARPES tight-binding simulation figures in the main text (Fig. 2 e, f) and in the supplementary (Supp. Fig. 4 l, o) to disordered average quantities, according to the discussion above. In SUPP. FIG. 3, we also added a panel showing the raw spectral function at a constant energy, the laplacian of which is showed in Fig. 2 f.

In my opinion there exist only the aTSS and there is no reason to invoke anything else. The states that they report do not exist in topologically trivial material where all back-scattering channels are allowed.

We completely agree with the referee in their first sentence. We do expect Brillouin-zone repetitions to exist in topological trivial amorphous materials. It is important to note that amorphous band-structures can exist even without protection by backscattering from topological properties. This was already pointed out by Edwards [1, 2], provided electrons scatter with correlated disorder. This is also mentioned in passing by Thouless in his review for Weaire-Thorpe models of the type we use in this work, see last paragraph of section 2.1 in [11]. We believe this is a key point of misunderstanding between us and the referee. To reiterate, bulk amorphous band structures also show Brillouin-zone-like repetitions, as seen in Fig. S4l and o.

Figure 2: Comparison between the figures shown in the previous version of the manuscript and the new version. The previous figures (left column), especially for Fig. 2e, were precise enough to see that the states that appear at commensurate momenta are copies of the central dispersive bands. Multiple copies appear since many different local configurations are realized in a single numerical sample. However, this fine structure averages out in the experiment due to the size of the sample and to the resolution of ARPES. Averaging over 100 disorder realizations (right column) allows us to reproduce this effect.

At present the authors have in their hands a good experimental proof of dispersive surface states in topological insulator. A paper based on such a basic, simple fact is something that I could support for publication. Gladly support, taking into account the amount of experimental work invested here. Of course that would require changing title, abstract, intro and conclusions (basically entire packaging of the experimental data). It would also require a serious conversation with Nature Comm. Editors, whether or not they see such an article in their journal.

We are glad to read that the referee has a positive opinion of our experimental data. We hope that with the above explanations our work is understood as direct experimental evidence of the local-order effects on electronic structure. These results in Brillouin-zone-like repetitions which have not been observed before.

Of course, on the theory level the manuscript is quite incremental (see e.g. Remark 2A); it would be for instance desirable if the theory can explain the “mysterious” k_z dependence in Fig.3e; but perhaps there is some substantial innovation on the experimental side that would nevertheless warrant the publication in Nat.Comm. It could be perhaps acceptable to put forward the hypothesis of general aSS in a Discussion section, if authors wish to do so. For instance something along the lines: “Here we show that well defined aTSS exist in TI, but perhaps they are more general phenomenon, which is exciting avenue for further research”

In our answer to Remark A we point out that most theory work in amorphous topological matter concerns models of uncorrelated disorder, which do not lead to Brillouin-zone repetitions. Hence, it is only with the theory presented in this paper that we can explain our findings. Specifically, the combination of Edwards theory and topological surface states is new to this work (Fig. 4d and S6), as well as the demonstration that tight-binding models, topological or not, with a characteristic real-space structure show Brillouin-zone-like repetitions (Fig. 2d and S4).

Our theory generally underlines the relation between the local order in real space and the long-range structure of the eigenstates in momentum space. In this sense, it indicates that the observation of dispersive states under ARPES even at large momenta may not be specific to topological boundary states.

Changes in the main text: Following the referee’s suggestion, we have reformulated the beginning of the discussion section to better emphasize that the different models we explored illustrate the general relation between local order and long-range structure in momentum space.

Actually, at present the Discussion section is actually quite puzzling, its second paragraph (about Fig.3d-e) gives impression that k_z momentum is not a good quantum number (because translational symmetry is lost) which agrees well with basic solid-state-textbook intuition, but is against what authors claim about BLZ for the in-plane momenta. Then a cautious reader asks himself a question: if an in-plane momentum supports BLZ and the perpendicular k_z host “subtle effects” (mixing of various k_z ??) then is the amorphous phase really spherically invariant, as it is emphasized in Fig.2?

Fig. 2 demonstrates cylindrically symmetric (not-spherically symmetric) states, as shown by the orange states in Fig. 3c. Due to the presence of the surface, the in-plane momentum dependence is expected to be very different from out-of-plane. The states are then probed along k_z using photon energy in Fig. 3d and 3e. Unlike the in-plane ring structure, the apparent k_z dependence in Fig. 3e does not correlate with any feature in the structure factor. Importantly, it also cannot be explained by a crystalline structure, because it does not correlate with the observed k_z range of the effect. Hence it is reported as an anomaly.

It is also not surprising that k_z is more subtle to interpret. This is also true for crystalline band structures. Higher photon energies eject electrons from deeper in the bulk. As they are ejected from the material can scatter more often than ejected surface electrons. This is at the heart of common assumptions in ARPES which we do not try to dive into. We expect this to be relevant too for glassy systems. We do not attempt to understand the subtleties of k_z physics here. We do try to account faithfully the differences we see with crystals.

Remark1A: Please note that Supp. Fig.4 in authors own manuscript shows a case of amorphous solid without SS, namely Se-rich sample. In my hypothesis, of the aTSS only, the large amount of Se (element with smaller spin-orbit coupling) suppress the gap inversion and thus the topological protection against back-scattering.

As explained in more details earlier, we do not claim that all amorphous systems should host topological surface states. It is indeed likely that Se-enrichment suppresses the gap inversion since the ARPES shows no topological state. This phenomenology would also be expected in a crystal.

Remark2A: the angle integrated ARPES is a local probe (I deduce it from Fourier transform’s definition) so it is not at all surprising that averaged core levels match each other. It is not a discovery, rather a necessary cross-check.

We believe the Referee may be referring to our comment about the right panels of Fig. 3d and e, in which the momentum integrated spectra look similar between amorphous and crystalline state or perhaps Fig. 3a and b when looking at deeper valence states. We would like to clarify a few points regarding this response. Firstly, ARPES

Figure 3: Bi 3d core level dispersion in ARPES

is a local probe in that band structure features can reveal features of local potentials, but it globally averages over many unit cells due to relatively large length-scale of the beam spot. Secondly, the referee mentions "core levels". Our paper does not mention core levels in the main, the deepest states we refer to are no more than 5 eV below the Fermi surface. Typically, in spectroscopy, core level correspond to high binding energy states far below the valence states and do not disperse (in ARPES they are flat lines, see response figure 3). It can be clearly seen in Fig. 3a, that these states are strongly dispersive in the crystalline phase, therefore correspond to the valence states. It is not trivial that the angle integrate spectra of the crystalline and amorphous phase should produce similar profiles. The local environment strongly influences the dispersion of these states (again evident by the crystalline example in Fig. 3a and is the reason why XPS is used as a technique to determine different bonding environments of the surface) and the symmetry of states relative to the experiment geometry can strongly affect the photoemission matrix elements. The fact that the integrated spectra appear similar suggests that many of the local parameters that define the amorphous and crystalline systems are similar, despite the global averaging effect of observing over many unit cells. Therefore we emphasize the point that the angle integrated spectra are similar as proof of similar local environments and order. This local order is the reason we observe momentum space structure and repetitions, the main claim of the paper, so we feel this point is important. We would like to point out that we agree with the referee regarding the point that the core levels (deep in binding energy) of the crystal and amorphous system should match and is not a discovery.

Changes in the main text: We can add a sentence to the main text regarding these figure panels if the referee deems it helpful, otherwise we will only address it in this response.

B. In their reply authors made an attempt to go beyond the theory from early 1960' and computed full Born

approximation, fBA, which was indeed rewarding for them as they were able to generate Fig. 4d. I am happy to inspire something that made authors excited enough to call it “tantalizing”. However in doing that the authors put on the center stage their amorphous carbon calculations. In my previous review I was rather lax about this sudden, surprising appearance of carbon allotrope in the manuscript where no experiment on carbon is reported. Now I see that this was my mistake.

We hope we have clarified in this response why we discuss amorphous graphene. It is there to corroborate that local order can lead to Brillouin zone repetitions.

I think authors should seriously re-consider removing those theoretical results on carbon. First and foremost: if one makes experiment on Bi₂Se₂ then he makes ALL computer simulations on Bi₂Se₃. There must be no ambiguity here, there is no circumstance/exception/explanation that would justify comparing any experimental finding in Bi₂Se₃ with numerical results on amorphous carbon (or mono-atomic lattice as the carbon is sometimes called). Reading present text I was sometimes puzzled if the experimental plot is not compared with carbon numerics, so let me re-iterate: this must not be done. (see Remark1B below)

We are willing to take the recommendation seriously and remove amorphous graphene. However, we still believe it serves as pedagogical example that supports the generality of the effect we observe. Since amorphous graphene exists and it is locally ordered, it seemed a natural pedagogical example to present and apply our theory to. It indeed shows Brillouin-zone-like repetitions (see Fig. S5). If the referee considers that it adds confusion, we are willing to remove it.

The logic of authors seem to be: look we have the same aSS states also in an amorphous carbon. So this must be the generic effect.

Respectfully, this is not our logic, but a short reflection of the referee’s opinion of our logic. We do claim that whenever short-range order in real space exists, there is an associated momentum scale. This causes Brillouin-zone-like repetitions in topological surface states of a-Bi₂Se₃. The same mechanism be at play for bulk states in amorphous carbon. We also point out that there are no a-SS in amorphous graphene: it is a single sheet, and in this sense all states we discuss are bulk states.

However, the authors do not have a full many-body solution for neither material. Tight-binding is one of the crudest approximation in solid state physics, it is even poorer than single-particle mean-field, like DFT. The most beautiful aspect of TSS is that the topological protection switches-off several relevant electron-impurity and electron-electron back-scattering terms – in Bi₂Se₃ this ultimately leads to the observed aTSS. In the carbon allotrope one can obtain surface state on a single particle level, but there is no reason that it will survive in a real material with all interactions included, only experiment can prove it. The authors performed the full Born approximation (fBA) calculation which is OK (actually nearly fBA because there is a strange ad-hoc alpha coefficient in the denominator in Eq. 4 in SM). But from Wilson-type RG work on Anderson localization we know that in 2D the localization appears only in RG (it is actually marginal term), to get it one thus need a parquet re-summation in perturbative language. The added-by-hand coefficient alpha, which appears where the vertex correction would have been, is not that innocent at all. To conclude: if authors would like to have evidence of the general aSS without experiment, but only based on the theory then this must be a theory with exact solution of a full many body problem. Only then there would be no loophole in the reasoning. This is probably even more difficult than the experimental evidence. In here the authors do not even study how the topological protection modifies their fBA. Hence their argument is massively insufficient.

We thank the referee for this analysis. The referee discusses several degrees of approximation and tools one can use to study disordered systems, each one with its own benefits and drawbacks. An important point to consider that *independent of topological properties*, one expects to find Brillouin zone repetitions for correlated disorder under quite general arguments, i.e. the existence of a typical real-space scale, or equivalently, a structure factor with well defined peaks [1, 2, 3]. This result does not depend on the way one attempts to solve the full many-body solution of the problem, provided this length scale is well defined. We have chosen two ways of solving the problem: based on the Born approximation and tight-binding models (DFT can only access ≈ 100 of sites for amorphous materials, and the limitations and successes of tight-binding theory for amorphous materials are also well-known since the 70's [12]). We agree that these are by no means the full many-body solution, but they are complementary and capture interesting aspects of the experimental data. We are also actively avoiding overselling of our result by invoking that our findings are magically a consequence of topological properties. As our graphene example shows, the explanation of our data has a much simpler, and beautiful origin, not directly observed in amorphous matter (see answer to Referee 2). We believe our approximations are sufficient to show that whenever short-range order in real space exists, there is an associated momentum scale. As an added plus, both recover features observed

in experiment. This causes Brillouin-zone-like repetitions, in topological surface states of α -Bi₂Se₃, or the bulk dispersion of amorphous graphene. It is hard to imagine that this basic fact about the problem can change by the level of approximation used to solve the disorder problem.

Remark1B: The fact that two materials host Dirac states does not make them equal, not even in a reciprocal space. In Bi₂Se₃ the Dirac states are due to spin-orbit interaction and are limited to in-gap energy range while in graphene they are in a much broader energy range and due to an underlying hexagonal crystal structure (and there is no spin-orbit coupling on an element as light as carbon). Even if we only build an effective theory in reciprocal space, in Bi₂Se₃ one has to take carefully into account the spin conservation in scattering which severely limits scattering phase space. One could continue long with such a list of differences... Here it is even worse: the authors pertain to study the role of variation of atomic positions. And on this microscopic level Bi₂Se₃ is completely different than monoatomic carbon! The analogy (if any) completely breaks down.

We believe this comment exemplifies the misunderstanding between us and the referee: as we discussed in our answer above, the Dirac nature of graphene electrons is not relevant to Brillouin-zone-repetitions. Provided a placement of atoms whose structure factor retains peaks, Brillouin-zone repetitions are expected to occur.

References

- [1] Edwards, S. F. The electronic structure of disordered systems. The Philosophical Magazine: A Journal of Theoretical Experimental and Applied Physics **6**, 617–638 (1961).
- [2] Edwards, S. F. & Mott, N. F. The electronic structure of liquid metals. Proceedings of the Royal Society of London. Series A. Mathematical and Physical Sciences **267**, 518–540 (1962).
- [3] Hafner, J. & Krajci, M. Electronic structure of quasicrystalline Al-Zn-Mg alloys and related crystalline, amorphous, and liquid phases. Physical Review B **47**, 11795–11809 (1993).

- [4] Zachariasen, W. H. The atomic arrangement in glass. Journal of the American Chemical Society **54**, 3841–3851 (1932).
- [5] Mitchell, N. P., Nash, L. M., Hexner, D., Turner, A. M. & Irvine, W. T. M. Amorphous topological insulators constructed from random point sets. Nature Physics **14** (2018).
- [6] Marsal, Q., Varjas, D. & Grushin, A. G. Topological weaire–thorpe models of amorphous matter. Proceedings of the National Academy of Sciences (2020). URL <https://www.pnas.org/content/early/2020/11/17/2007384117>.
- [7] Corbae, P., Hannukainen, J. D., Marsal, Q., Muñoz-Segovia, D. & Grushin, A. G. Amorphous topological matter: Theory and experiment. Europhysics Letters **142**, 16001 (2023). URL <https://dx.doi.org/10.1209/0295-5075/acc2e2>.
- [8] Agarwala, A. & Shenoy, V. B. Topological insulators in amorphous systems. Phys. Rev. Lett. **118**, 236402 (2017).
- [9] Toh, C.-T. et al. Synthesis and properties of free-standing monolayer amorphous carbon. Nature **577**, 199–203 (2020).
- [10] Weiße, A., Wellein, G., Alvermann, A. & Fehske, H. The kernel polynomial method. Rev. Mod. Phys. **78**, 275–306 (2006). URL <https://link.aps.org/doi/10.1103/RevModPhys.78.275>.
- [11] Thouless, D. Electrons in disordered systems and the theory of localization. Physics Reports **13**, 93–142 (1974).
- [12] Weaire, D. & Thorpe, M. F. Electronic properties of an amorphous solid. i. a simple tight-binding theory. Phys. Rev. B **4**, 2508–2520 (1971). URL <https://link.aps.org/doi/10.1103/PhysRevB.4.2508>.

REVIEWERS' COMMENTS

Reviewer #2 (Remarks to the Author):

The revised manuscript seems to be far less misleading in terms of the true novelty of this work over the previous reports. I agree on the point that in the case of quasicrystals, there remains a long-range rotational symmetry so that the sharp structure factors can be clearly observed in diffractions. The observation of band replica even in the absence of such long-range order may be a novel finding, and more importantly, I totally agree on the point that this fact is relatively not well known in the community even though it is fundamentally important to understand locally ordered materials.

However, let me get this point straight. Strictly speaking, the observation of band modulations by the effect of a local order or the ring-shape "quasi-Brillouin zone" is not completely new, for example, the same physics has been discussed in terms of the papers by Edwards, see Fig. 4a,b, Phys. Rev. Lett. 107, 136402 (2011). The authors might argue that there are some differences between liquid metals and amorphous solids. However, what is important for the physics of Brillouin zone repetitions here is whether there is the presence of a local order (which is the same for liquid metals and amorphous solids) rather than whether the constituent atoms are moving or not (that's why theoretical works by Edwards were basically concerned with liquid metals).

Nevertheless, I feel this physics is relatively not well known than it deserves as exactly pointed out by the authors. This is another experimental demonstration of this physics with amorphous solids, which will contribute to stimulating new thinkings in the community.

If the authors appropriately revise their abstract and introduction to differentiate more clearly what can be said completely new in this work from what has been already known and discussed before, I will be happy to support publication of this work in Nature Communications.